# Rule-Based Rating and Selection of LLM Training Data

## Abstract

The quality of training data is crucial for the performance of large language models (LLMs). There are recent studies utilizing LLMs to rate and select data based on scores from a small set of human-designed metrics (rules). However, existing rule-based methods often overly rely on human heuristics, lack robust metrics for rule evaluation, and exhibit limited adaptability to new tasks. In our work, we propose a novel rule-based framework that leverages the orthogonality of score vectors corresponding to rules as a unique metric for rule evaluation. Our method employs an automated pipeline that first uses LLMs to generate a diverse set of rules, covering a wide range of rating aspects. It then rates a batch of data according to these rules and applies the determinantal point process (DPP) from random matrix theory to select the most orthogonal score vectors, effectively isolating a subset of independent rules. Then these rules are applied to rate all data and samples with the highest average scores are selected for further downstream tasks such as LLM training. We validate our method through two experimental setups: 1) comparison against ground truth ratings and 2) benchmarking LLMs trained with the selected data. Our extensive experiments span various settings, including general pre-training and domain-specific fine-tuning in fields such as IMDB, Medical, Math, and Code. The results show that our DPP rule-based rating method consistently outperforms other methods, such as rating without rules, uniform sampling, importance resampling, and QuRating, in terms of both rating accuracy and model performance.

## 1 Introduction

Large language models (LLMs) have been widely utilized across a diverse range of applications. Pretraining and fine-tuning these models typically require large and diverse datasets. Studies have found that data quality is critical for training good LLMs (Brown, 2020; Chowdhery et al., 2023; Du et al., 2022; Dubey et al., 2024; Wenzek et al., 2019). For instance, Meta's LIMA paper (Zhou et al., 2024) demonstrated that using only 1000 carefully curated data samples can achieve better performance than using the original 50k samples. Similar phenomena have been observed in other studies where selecting a subset of high-quality datasets increases the training convergence and model performance (Cao et al., 2023; Hsieh et al., 2023; Xie et al., 2024; Sachdeva et al., 2024; Zhang et al., 2023; Javaheripi et al., 2023).

Recent studies now adopt an approach that employs LLM-as-a-judge to grade data quality according to a set of designed metrics (which we call rules) (Yuan et al., 2024; Wettig et al., 2024; Bai et al., 2022; Mu et al.). For example, Wettig et al. (2024) rates the pre-training data using LLMs according to four predefined rules. RedPajama (Together AI, 2023) is continuously developing a pool of rules for users to select from, which currently contains over 40 basic criteria that LLM data should satisfy. On specific aspects such as safety, Constitutional AI (Bai et al., 2022) proposed their "constitution"—a set of standard safety criteria—to generate safe synthetic data, and in Huang et al. (2024) they have developed 133 rules. Most recently, OpenAI's Rule-based Rewarding (Mu et al.) proposed 21 general safety rules and injected them into the RLHF (reinforcement learning with human feedback) process. This rule-based rating provides greater explainability of data quality and breaks down the challenge of assigning a data point one overall quality score into a simpler task of giving several rule-specific scores. Evidence suggests that this fine-grained approach also yields more accurate rating outcomes (Yuan et al., 2024; Wettig et al., 2024; Bai et al., 2022; Mu et al.).

Nonetheless, there are several critical problems and challenges. First, designing an effective set of rules is quite difficult, a fact acknowledged by most of the papers above. Current designs in these papers all rely heavily on human heuristics and are sometimes too broad for effective rating. Second, as far as we know, the metrics to evaluate rules are lacking, and there has been no systematic exploration of how different rule choices and sizes impact the outcomes. In previous experiments Bai et al. (2022); Wettig et al. (2024); Together AI (2023), a subset of rules is selected (typically randomly) during the rating process. This selection can significantly influence the rating results and consequently the quality of the sampled data. Furthermore, the utility and impact of rules can vary significantly, and some rules are strongly correlated with each other, as highlighted in Wettig et al. (2024), which introduces redundancy and bias into the rating procedure. Therefore, a critical question arises: with a "constitution" (a pool of rules) in hand, exactly which "laws" (a subset of task-related rules) should be applied to a specific task? Random selection as in Bai et al. (2022) may not be the optimal strategy. A third major drawback is the inflexibility of these rules; they are often designed for specific settings, such as pre-training or safety tasks, not generally applicable across different settings.

In our work, we aim to address these challenges. First, we leverage an LLM (GPT-4 Achiam et al. (2023)) for automatic rule generation, where we include the descriptions of the task and source dataset in the prompt. At this stage, some generated rules are found to be repetitive or redundant, similar to the issues in human-designed rules. Our strategy is to first generate a comprehensive set of rules to ensure a broad diversity that covers various aspects we seek to evaluate in the data, and then filter out repetitive rules. Hence the second step of our approach is to select a subset of rules that are relatively uncorrelated/independent. This is achieved by first using the rules to rate a batch of data, creating one score vector for each rule, and then assessing the independence of rule subsets through the overall orthogonality of their corresponding score vectors. We propose the formula in Section 3 to measure the orthogonality and use determinant point process (DPP) sampling (Macchi, 1975; Borodin & Olshanski, 2000) to identify a subset of independent rules. Once the rules are determined, the third step is to use them to rate all source data and select the high-quality ones. Combining these steps of rule generation, rule-based rating, rule selection by DPP, and data selection, our method establishes a fully automated framework for rule-based data selection (illustrated in Figure 1). Notably, we are the first to introduce the mathematical rule evaluation metric, based on the orthogonality of their score vectors. Moreover, our pipeline does not need human intervention in designing and selecting rules at all. For every new task, one can use the pipeline to get a set of high-quality, task-specific rules at low cost. These address the challenges of existing methods mentioned above. Another advantage of our method is its natural extension, allowing for customization such as re-weighting particular rules, and this is only feasible when the selected rules are relatively "orthonormal".

Note that our data selection methodology is highly versatile and applicable to a variety of scenarios, including LLM pre-training, fine-tuning on specific domains, RLHF preference data, etc. We conduct experiments to cover a range of tasks and datasets, including general pre-training data and domain-specific data in four domains: IMDB, Medical, Math, and Code.[1] We show that our rule-based data selection typically yields more accurate rating results, thereby enhancing data quality and leading to better performance of the LLM trained with the data. Here is a list of the main contributions of our work:

1. **Rule-free vs. Rule-based Rating.** Our systematic experiments demonstrate that fine-grained rule-based rating outperforms rule-free methods, producing more precise data quality assessments, leading to improved benchmark performance of LLMs.
2. **Rule Evaluation Metric:** We introduce a novel rule evaluation metric designed to promote low correlation and high diversity among rules. We propose the method of using DPP on task-aware rule-rating vectors to select a subset of independent rules.
3. **Automated Rule-based Rating and Selection Pipeline.** We confirm that LLMs are effective rule generators, eliminating the need for manual rule crafting. Our automated pipeline generates the rules, selects the rules, and then chooses data according to rule-based ratings. This entire process operates independently of human heuristics and is free from human biases.
4. **Cross-Domain and Cross-Model.** We validate our method through two approaches: A) comparison with ground truth ratings, and B) training LLMs with selected data and assessing perfor-

---

[1]Code of our experiments: https://anonymous.4open.science/r/DataSelection-F118/

mance across various benchmarks. Our experiments span multiple models, including Pythia-1B and Llama3-8B (fine-tuned with LoRA), and cover diverse domains such as IMDB, Medical, Math, and Code, confirming the versatility and model independence of our approach.

## 2 RELATED WORK

**LLM data selection.** There are different genres of data selection approaches for LLMs. Basic filterings, such as setting thresholds on word lengths, are used in many studies to eliminate low-quality data (Soldaini et al., 2024; Wenzek et al., 2019; Raffel et al., 2020; Conneau & Lample, 2019; Penedo et al., 2023; Laurençon et al., 2022). Fuzzy deduplication is another approach which removes repetitive or similar data samples (Allamanis, 2019; Lee et al., 2021; Abbas et al., 2023; Gao et al., 2020; Jiang et al., 2022). Another method is "heuristic classification", selecting data based on a predefined quality score, typically measured by similarity to formal sources such as Wikipedia or other human-generated, high-quality datasets (Brown, 2020; Touvron et al., 2023; Chowdhery et al., 2023; Du et al., 2022; Gao et al., 2020; Wenzek et al., 2019). In contrast to this, directly querying LLMs to rate data and use the scores as the quality indicator has become a standard practice in many studies (Li et al., 2023a; Chen et al., 2023; Bai et al., 2022; Wettig et al., 2024; Yuan et al., 2024; Dubois et al., 2024; Li et al., 2023b; Fernandes et al., 2023).

**Rule-based rating.** There are studies adopting a more fine-grained approach to data quality, distilling it into a finite set of metrics which we refer to as "rules". For instance, RedPajama (Together AI, 2023) provides over 40 quality rules that serve as basic quality metrics for the users to choose from. More pertinent to our research, there are papers that apply this rule-based idea to rate LLM data. For example, Yuan et al. (2024) assigns a score out of 5 to each data point, awarding 1 point for each of the 5 predefined criteria met. In Wettig et al. (2024), the authors designed four general rules to rate and select data for LLM pre-training. (Sun et al., 2024) proposed 16 human-crafted rules to evaluate the desirable quality of response data. The rule-based approach is also utilized in more targeted applications, such as ensuring data safety. Constitutional AI designed 16 general safety critique rules to revise synthetic data, enhancing data safety (Bai et al., 2022). This revision process involves iterative steps where a random subset of rules from the "constitution" (the entire set of rules) is applied. Additionally, in Mu et al., the score generated by an LLM grader according to a set of 21 safety rules is integrated directly into the RLHF process as an additional reward. In Wang et al. (2024), they design a composite reward model in RLHF, trained using rule-based ratings. As noted earlier in the introduction, the rules employed in the literature exhibit several critical issues. They often depend heavily on human heuristics for design, lack robust rule evaluation metrics and exploration of rule sizes, and demonstrate limited versatility for new tasks or for customization. Our goal is to address these challenges using our proposed framework.

## 3 METHODOLOGY

### 3.1 DEFINITIONS AND NOTATIONS

We introduce the definitions of the primary objects considered in our method:

- $R$: the total number of available rules.
- $r$: the number of selected rules, using a specified rule selection method.
- $\mathcal{D}$: the set of all data samples, with its size denoted by $N \overset{\text{def}}{=} |\mathcal{D}|$.
- $\mathcal{B} \subseteq \mathcal{D}$: a batch of data samples, randomly selected for evaluating the correlation of rules during the rule selection step, with its size denoted by $n \overset{\text{def}}{=} |\mathcal{B}|$.
- $\boldsymbol{S} \in \mathbb{R}^{n \times R}$: the rating matrix $\boldsymbol{S}$ where each entry $S_{i,j}$ represents the score of the $i$-th data sample according to the $j$-th rule and is constrained to the interval $[0, 1]$.
- $\bar{\boldsymbol{S}} \in \mathbb{R}^{n \times r}$: a submatrix of $\boldsymbol{S}$ consisting of the $r$ selected columns from $\boldsymbol{S}$, corresponding to the $r$ selected rules.

**Measure orthogonality:** We propose a metric for selecting rules based on the orthogonality of score vectors. Here we introduce a mathematical definition to quantify the orthogonality or correlation of

a set of score vectors. Given a score matrix $\bar{\boldsymbol{S}} \in \mathbb{R}^{n \times r}$ such that the columns are the score vectors of dimensions $n$. We begin by computing the covariance matrix $Cov(\bar{\boldsymbol{S}})$ for the columns of $\bar{\boldsymbol{S}}$, whose entries are defined by

$$Cov(\bar{\boldsymbol{S}})_{i,j} \stackrel{\text{def}}{=} \frac{1}{n} \sum_{k=1}^{n} (S_{k,i} - \mu_i)(S_{k,j} - \mu_j), \qquad 1 \le i, j \le r,$$

where each $\mu_i \stackrel{\text{def}}{=} \frac{1}{n} \sum_{k=1}^{n} S_{k,i}$ is the sample mean for rule $i$. Then define the sample correlation matrix as $Corr(\bar{\boldsymbol{S}}) \in \mathbb{R}^{r \times r}$ where

$$Corr(\bar{\boldsymbol{S}})_{i,j} \stackrel{\text{def}}{=} \frac{Cov(\bar{\boldsymbol{S}})_{i,j}}{\sqrt{Cov(\bar{\boldsymbol{S}})_{i,i} \cdot Cov(\bar{\boldsymbol{S}})_{j,j}}}, \qquad 1 \le i, j \le r.$$

Commonly used libraries such as `Numpy` provide straightforward functions to compute the correlation matrix. We introduce the concept of *rule correlation*, which quantifies the degree of correlation/dependence for a given rating submatrix $\bar{\boldsymbol{S}}$, defined as follows:

$$\rho(\bar{\boldsymbol{S}}) \stackrel{\text{def}}{=} \frac{1}{r} \|Corr(\bar{\boldsymbol{S}}) - \boldsymbol{I}_r\|_F = \frac{1}{r} \sqrt{\sum_{i \ne j} Corr(\bar{\boldsymbol{S}})_{i,j}^2}. \tag{1}$$

Here $\boldsymbol{I}_r \in \mathbb{R}^{r \times r}$ is the identity matrix, and $\|\cdot\|_F$ represents the Frobenius norm. This metric quantifies how much the columns of $\bar{\boldsymbol{S}}$ deviate from orthogonality, by measuring the deviation of its correlation matrix from the identity matrix. The second equality in 1 provides another intuitive understanding: $\rho(\bar{\boldsymbol{S}})$ essentially aggregates the correlations of all pairwise correlations of rules $(i, j)$ for $i \ne j$.

## 3.2 DETERMINANTAL POINT PROCESS (DPP)

The optimal solution to this mathematical problem of selecting the most orthogonal subset of a set of vectors is NP-hard (Civril & Magdon-Ismail, 2007; Kulesza et al., 2012) but we use DPP sampling to provide a relatively good solution. The determinant point process (DPP) is a probabilistic model that describes the likelihood of selecting diverse subsets from a larger set (Macchi, 1975; Borodin & Olshanski, 2000). Mathematically, a DPP is defined by a kernel matrix that describes the similarities between elements in a set. The probability of selecting a particular subset is proportional to the determinant of the corresponding submatrix of this kernel matrix. Intuitively, subsets with highly similar items (leading to higher correlation in the submatrix) have smaller determinants and are thus less likely to be chosen.

**DPP Definitions.** Given a discrete ground set $\mathcal{Y}$, without loss of generality we let $\mathcal{Y} = \{1, 2, \ldots, R\}$, a (discrete) DPP defines a probability measure over $2^{\mathcal{Y}}$, the power set of $\mathcal{Y}$. Let $Y$ be a randomly chosen subset. Then for any subset $A \subseteq \mathcal{Y}$, the probability of $A$ being chosen by a DPP is given by:

$$\mathbb{P}(A \subseteq Y) = \det(\boldsymbol{K}_A)$$

where $\boldsymbol{K} \in \mathbb{R}^{R \times R}$ is a real positive-semidefinite matrix called the *kernel matrix* and $\boldsymbol{K}_A \stackrel{\text{def}}{=} [\boldsymbol{K}]_{i,j \in A}$ is the submatrix of $\boldsymbol{K}$ indexed by elements in $A$.

**Kernel Matrix.** Each entry $K_{ij}$ in the kernel matrix $\boldsymbol{K}$ describes the similarity between elements $i$ and $j$ in $\mathcal{Y}$. For our purpose of selecting orthogonal rules, we will define $\boldsymbol{K}$ as the Gram matrix of the score vectors: $\boldsymbol{K} \stackrel{\text{def}}{=} \boldsymbol{S}^\top \boldsymbol{S}$.

**DPP Sampling.** To sample a diverse subset using DPP, there are several existing algorithms (Hough et al., 2006; Kulesza et al., 2012; Tremblay et al., 2018) and the Python library `DPPy` (Gautier et al., 2019) implements some of them. The computation of the DPP sampling primarily hinges on the overhead of computing the inner product kernel matrix $\boldsymbol{K}$ and its eigendecomposition. In our case, $\boldsymbol{K} \in \mathbb{R}^{R \times R}$ and hence it requires $O(R^3)$ time, where $R$ is the number of all rules. Nonetheless, we set $R = 50$ in our experiments, therefore our DPP rule selection algorithm is extremely fast (typically within 0.1 seconds). Further details about DPP sampling algorithms and their time complexities can be found in Appendix A.3.

## 3.3 DPP RULE-BASED RATING ALGORITHM

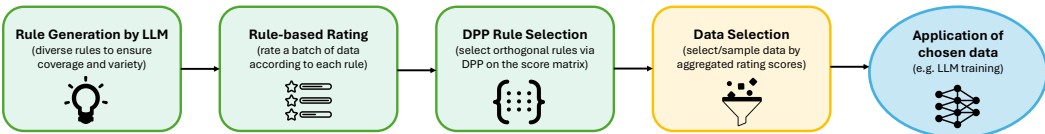

Figure 1: Pipeline for rule-based data rating and selection. Step 1. Use LLM to generate a comprehensive set of $R$ rules. Step 2. Rate a batch of $n$ data according to $R$ rules and form the score matrix $\boldsymbol{S} \in \mathbb{R}^{n \times R}$. Step 3. Select $r$ rules that correspond to the columns sampled by the DPP in the score matrix. Step 4. Rate the full dataset using the $r$ selected rules and (stochastically) select data with the highest averaged ratings. Step 5. Application of chosen data on downstream tasks, such as for LLM training.

The pipeline of our rule-based data selection method is illustrated in Figure 1 and comprises the following steps:

**Step 1. Rule generation.** We query GPT-4 to generate $R$ rules. In the prompt, we include the goal, the description of the source data, and the description of the downstream task to help GPT-4 generate relevant task-related rules.

**Step 2. Rule-based rating**: Recall the definitions in Section 3.1. We employ LLM, particularly Llama3-8B-Instruct (AI@Meta, 2024), to rate the batch data $\mathcal{B}$ according to $R$ rules, resulting in the matrix $\boldsymbol{S} \in \mathbb{R}^{n \times R}$.

**Step 3. Rule selection using DPP**: From $\boldsymbol{S}$, we aim to select $r$ relatively independent columns using a DPP, forming the submatrix $\bar{\boldsymbol{S}} \in \mathbb{R}^{n \times r}$. We define the kernel matrix of DPP as follows:

$$\boldsymbol{K} \overset{\text{def}}{=} \boldsymbol{S}^\top \boldsymbol{S} \in \mathbb{R}^{R \times R}, \tag{2}$$

where each entry $K_{i,j} = \langle S_i, S_j \rangle$ (each $S_i$ is the $i$-th column of $\boldsymbol{S}$), representing the similarity between rule $i$ and rule $j$. We then employ the DPP sampling algorithm to select $r$ indices from $\{1, 2, \ldots, R\}$, corresponding to the $r$ chosen rules.

Note that the cost of generating $R$ rules is negligible, requiring just a single GPT-4 query, and the cost of obtaining the rating matrix $\boldsymbol{S}$ can be managed by adjusting the batch size $n$. The motivation to select a fixed small number of $r$ rules is driven by the computational costs associated with using LLMs for data rating and the need to maintain a consistent dimensionality for explaining data quality. These practical considerations lead us to treat $r$ as a hyperparameter. Discussions on the optimal choices of $r$ are explored in Section 4 and Appendix A.7.5.

Another important remark is that, even with the same set of rules, they could have different correlations conditioned on a specific task or dataset. Therefore during DPP selection, instead of employing fixed representations such as semantic encodings—which result in static rule representations and selections across all tasks—we use *task-aware* score vectors to adaptively represent the rules. These vectors allow the entire pipeline to be customized for a particular downstream task.

**Step 4. Stochastic data selection**: We extend the rating process to cover all data samples using the selected $r$ rules, expanding the rating matrix $\bar{\boldsymbol{S}}$ from $n \times r$ to $N \times r$. We then aggregate these fine-grained ratings by averaging across the $r$ columns of $\bar{\boldsymbol{S}}$, resulting in a score vector $\boldsymbol{v} = [v_1, v_2, \ldots, v_N]$ that assigns a quality score to each of the $N$ samples.

Given the $N$ scores and a fixed budget of selecting $k$ samples for training, rather than choose the traditional top-$k$ approach, (selecting the $k$ highest scored samples), we adopt a stochastic sampling strategy, where we sample $k$ data points according to the distribution:

$$p(\boldsymbol{x}_i) = \frac{e^{v_i}}{\sum_{j=1}^{N} e^{v_i}} \tag{3}$$

for each data point $\boldsymbol{x}_i \in \mathcal{D}$. This stochastic data selection mechanism introduces greater diversity into the sampling process and is used in several other papers ((Wettig et al., 2024; Sachdeva et al., 2024)).

**Step 5.** Apply the selected data on given downstream tasks, such as for LLM pre-training and domain fine-tuning.

# 4 EVALUATION A: EVALUATING AGAINST GROUND TRUTH RATINGS

We evaluate our method in two ways: A. by comparing the rating results against the ground truth rating of the dataset. Smaller deviations from the ground truth scores indicate better performance. Specifically, we rely on pairwise comparisons generated by GPT-4 and apply the Bradley-Terry model (Bradley & Terry, 1952) to compute $n$ scores, treating them as the ground truth. B. by training an LLM (Llama3-8B) with the selected data and assessing its performance through both general and domain-specific benchmarks. In this section, we present the first set of experiments (corresponding to Evaluation A), while the second set of experiments (based on Evaluation B) is discussed in Section 5. The low cost of Evaluation A enables us to explore various aspects such as the rule-size scaling law, different rating schemes (pairwise vs. single), and the impact of model sizes (Llama3-8B and Llama3-70B). These experiments provide preliminary evaluations of our method.

## 4.1 EXPERIMENTS SETUP

**Datasets:** We consider two datasets: CommonCrawl (Common Crawl, 2024), containing raw web-crawled data, and IMDB (Maas et al., 2011), a dataset of 50K movie reviews, representing general and domain-specific settings, respectively. For each dataset, we collect the first 50 examples and apply a pairwise comparison scheme for data rating (prompt templates are available in Appendix A.6.6), which requires comparison on 2,450 ordered pairs.

**Ground truth scores:** Ground truth scores are generated as follows: we prompt GPT-4 to compare each pair of data samples $(i, j)$ and then reversing the comparison for $(j, i)$. We only keep the pairs where both comparisons are consistent, filtering out cases where GPT-4 performs poorly. After filtering, approximately 1000 comparisons remain for CommonCrawl and 1800 for IMDB. From these outcomes, we calculate scores for the 50 samples using the Bradley-Terry model (Bradley & Terry, 1952) (details can be found in Appendix A.6.1).

**Rating:** Now with the ground truth scores, we use our rule-based approach to rate the same data. For each rule $i \in \{1, 2, \ldots, R\}$ ($R = 50$), we employ Llama3-8B-Instruct as our comparison rater and similarly use the Bradley-Terry model to compute a score vector $S_i \in \mathbb{R}^n$ ($n$ is also 50 here), thereby forming the rating matrix $\boldsymbol{S} \in \mathbb{R}^{n \times R}$. Recall we denote $\bar{\boldsymbol{S}}$ as the submatrix of $\boldsymbol{S}$ containing $r$ columns indexed by the $r$ selected rules. To assess the rating results in $\bar{\boldsymbol{S}}$ against the ground truth, we compute the mean squared error (MSE):

$$\epsilon(\bar{\boldsymbol{S}}) \stackrel{\text{def}}{=} \frac{1}{n} \left\| \frac{1}{r} \sum_{j=1}^{r} \bar{S}_j - S_{GT} \right\|_2^2 \tag{4}$$

where $S_{GT} \in \mathbb{R}^n$ is the ground truth score vector and $\bar{S}_j$ is the $j$-th column of $\bar{\boldsymbol{S}}$. Furthermore, to establish comparative baselines, we implemented the same rating procedure (pairwise comparisons and score calculations via the Bradley-Terry model) using both the four designed rules in QuRating (see Wettig et al. (2024)) and a rule-free approach, referred to as the "NoRule" setting.

Our experiments in this section aim to address the following research questions: **(Q1)** Does greater rule diversity lead to more accurate ratings? **(Q2)** Does rule-based selection generally outperform rule-free methods? **(Q3)** How does our DPP-based rule selection compare to human-designed rules and ratings without rules? **(Q4)** Does DPP select better rules than randomly chosen ones? **(Q5)** How do different rating schemes and rater models impact the performance of our method?

## 4.2 RESULTS

**Correlation of $\rho(\bar{\boldsymbol{S}})$ and the MSE $\epsilon(\bar{\boldsymbol{S}})$ (answer to Q1).** For each $r \in \{1, 2, \ldots, 50\}$, we sample $\min\{10000, \binom{50}{r}\}$ sets of indices of size $r$, which are used to choose rules and form $\bar{\boldsymbol{S}}$. We then calculate its rule correlation $\rho(\bar{\boldsymbol{S}})$ and MSE $\epsilon(\bar{\boldsymbol{S}})$. We compute their Pearson correlation and observe positive values for both IMDB and CommonCrawl datasets (see Figures 2a and 2b). This confirms

that higher rule diversity is positively correlated with the accuracy of rating results. In other words, the correlation or redundancy of rules is positively correlated with the error $\epsilon(\bar{\boldsymbol{S}})$.

**Rule-based v.s. Rule-free (answer to Q2):** We sample $10^6$ possible rule subsets with size $r$ from all 50 rules and calculate the corresponding MSE, comparing it to the MSE from the NoRule setting. The results in Figures 2c and 2d demonstrate that using rule-based rating is mostly guaranteed to give better results than rating without rules, no matter applied to general data like CommonCrawl or domain-specific data like IMDB. When compared to QuRating MSE, the results show that QuRating is outperformed by most randomly selected rule subsets, highlighting the limitations of human-designed rules.

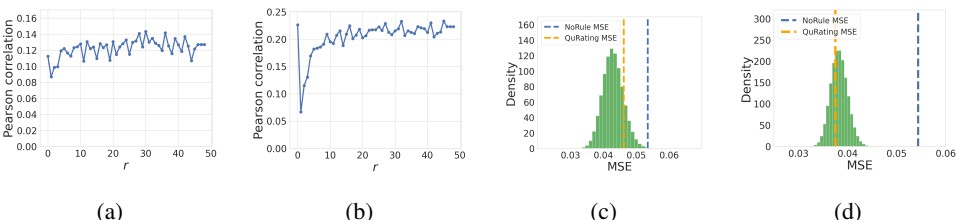

(a)      (b)      (c)      (d)

Figure 2: (a) and (b): Pearson correlation of the rule correlation $\rho(\bar{\boldsymbol{S}})$ and the MSE $\epsilon(\bar{\boldsymbol{S}})$ for IMDB and CommonCrawl datasets respectively. (c) and (d): Distribution of MSE across $10^6$ random rule subsets with size $r$, for IMDB and CommonCrawl datasets respectively, with vertical lines representing the MSE values of QuRating and NoRule.

**DPP v.s. QuRating v.s. NoRule (answer to Q3).** For each $r \in \{1, 2, \ldots, 50\}$, we use DPP to sample $r$ rules and conduct 100 trials. Then compare the averaged MSE against the MSEs from QuRating and NoRule, recording the winning rates of the DPP rules (see Figure 3a). For the IMDB dataset, we found that once $r$ reaches a certain threshold, DPP rules consistently achieve near-perfect winning rates against both NoRule and QuRating. Interestingly, for CommonCrawl, DPP underperforms QuRating when $r$ is too small or too large. This suggests that while QuRating rules are effective for general pre-training data, they lack the flexibility to adapt to other settings or domains.

**DPP rules v.s. Randomly selected rules (answer to Q4).** We compare DPP-selected rules with randomly selected rules of the same size $r$, evaluating both the rule correlation $\rho(\bar{\boldsymbol{S}})$ and MSE $\epsilon(\bar{\boldsymbol{S}})$ for their corresponding score submatrices $\bar{\boldsymbol{S}}$. The results show that DPP consistently produces rules with lower correlation and MSE, regardless of the value $r$ (see Figures 3b and 3c). Another key observation is that the MSE for DPP rules increases when $r$ is either too small or too large, with the optimal $r$ falling somewhere in the middle. This matches our intuition: when $r$ is too small, there are too few rules to achieve sufficient rating diversity, and when $r$ is too large, rule redundancy can negatively affect the rating outcomes. In fact, this motivates our selection of $r = 10$ in this paper.

**Variations in rating schemes and rater models (answer to Q5).** To verify that our method works across different rating schemes and rater models, we explored the following variations: *1. Pairwise v.s. individual rating.* While the pairwise ratings provide more reliable comparisons, individual rating requires only $O(n)$ computation. We observed similar results as in Section 4.2 (see Appendix A.6.3). Notably, individual ratings on the IMDB dataset showed a Pearson correlation between rule correlation $\rho(\bar{\boldsymbol{S}})$ and MSE $\epsilon(\bar{\boldsymbol{S}})$ of up to 0.6, and the winning rates show that DPP significantly outperforms both QuRating and the NoRule. *2. Llama3-8B v.s. Llama3-70B.* We tested the influence of rater model capability by switching to Llama3-70B (instruction-tuned version), using the individual rating scheme on IMDB. The results are similar to earlier and we also noted a high Pearson correlation (over 0.6) between rule correlation and MSE, along with a high winning rate of DPP compared to QuRating and NoRule. Furthermore, randomly selected rules perform significantly better than both QuRating and NoRule. See Appendix A.6.4 for further details.

## 5 EVALUATION B: DATA SELECTION FOR LLM FINE-TUNING

In this section, we follow the pipeline outlined in Section 3.3 and conduct experiments based on Evaluation B, where we train an LLM (Llama3-8B) using the selected data and assess its perfor-

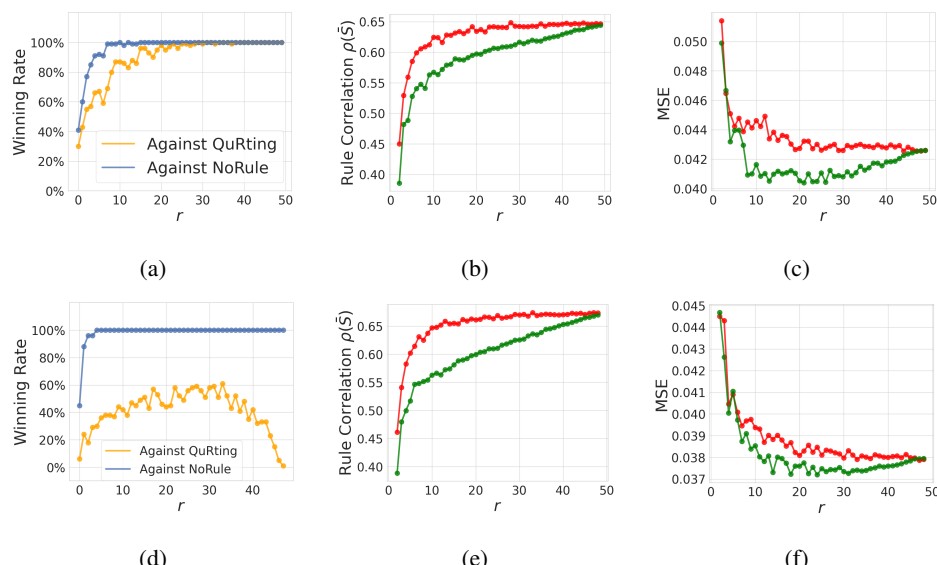

Figure 3: (a) Winning rate of DPP-selected rules compared to QuRating's four rules and the NoRule setting, based on MSE across 100 DPP trials. (b) Comparison of rule correlation between DPP-selected and randomly selected rules, averaged across 100 trials. (c) Comparison of MSE between DPP-selected and randomly selected rules, averaged across 100 trials. Plots (a), (b), and (c) display results for the IMDB dataset, while (d), (e), and (f) for the CommonCrawl dataset.

mance. This setup closely reflects real-world applications of LLM data selection. We benchmark our method against several baselines, such as uniform sampling, direct rating without rules, QuRating rules (Wettig et al., 2024), and DSIR (Xie et al., 2024) (a commonly used baseline for LLM data selection). We condcut experiments in this section to explore the following research questions: **(Q1)** How does data selected by rule-based methods enhance model fine-tuning compared to rule-free methods? **(Q2)** How do the rules generated by our automated framework compare to human-designed rules? **(Q3)** How does DPP rule selection perform compared to random rule selection?

## 5.1 EXPERIMENTS SETUP

**Evaluation Benchmarks.** To systematically evaluate the effectiveness of our framework, we use following benchmarks: For experiments on general continued pre-training, we utilize ARC-Easy (Yadav et al., 2019), ARC-Challenge (Yadav et al., 2019), Winogrande (Sakaguchi et al., 2021), MMLU(Hendrycks et al., 2020), and SST-2 (Socher et al., 2013). Then we employ domain-specific datasets to do fine-tuning: For IMDB, we use the IMDB sentiment analysis dataset (Maas et al., 2011). For Code, we use benchmarks for code generation, including HumanEval (Chen et al., 2021), MBPP (Austin et al., 2021), Multiple-py and Multiple-cpp (Cassano et al., 2022). For Math and Medical domains, we choose subsets from MMLU corresponding to Math subject and Medical subject respectively. More details about these benchmarks are summarized in Appendix A.7.4.

**Data Source.** SlimPajama is a large, deduplicated, multi-corpus open-source dataset specifically designed for training large language models (Cerebras Systems, 2023). We randomly sampled 1 million data points (around 1 billion tokens) from SlimPajama as our initial data source $\mathcal{D}$. From this pool, we employ our selection methods to choose data for training.

**Models.** We train Pythia-1B (Biderman et al., 2023) on general continued pre-training and domain fine-tuning for IMDB and Medical. We intentionally selected Pythia-1B because it is known to be pre-trained on the Pile dataset (Gao et al., 2020), making it a better choice than models that possibly included SlimPajama in their pre-training corpus. To validate the transferability of our framework across different LLMs, we train Llama3-8B (AI@Meta, 2024) with LoRA (Hu et al., 2021) for the Math and Code domains.

**Compared Methods.** We compare our method against the following baselines, including both rule-free and rule-based data selection methods. For rule-free methods, we have: *Uniform sampling*: select the data randomly, *No Rule:* prompt Llama3-8B-Instruction to individually rate the data without rules, and then apply the same sampling procedure as described in 3.3, *DSIR* (Xie et al., 2024): importance resampling of data that resemble a target dataset (we use Wikipedia as the target for the general continued pre-training and benchmark test datasets for the domain fine-tuning). For rule-based methods, we include: *QuRating* (Wettig et al., 2024): data rating and selection using four human-designed rules, and *GPT-Uncorrelated*: Directly prompting GPT-4 to generate 10 uncorrelated rules for data rating and selection. We have comparison against more baseline methods in Appendix A.7.2.

For our automated rule-based selection algorithm, we set $R = 50$, as in Section 4, and select $r = 10$ for rule selection. As inferences of LLM are a lot less computing-consuming than model training, we set $n = 10^4$ in all our experiments. The choice of $r$ as a hyperparameter is based on experimental observations from Section 4, where very small or very large values of $r$ did not yield optimal results. Discussion and exploration of different values for $r$ are provided in Appendix A.7.5, and details of the rule generation prompts, rating prompts, generated and selected rules are in Appendix A.7.11. To demonstrate the effectiveness of our automated orthogonal-rule-based selection algorithm, we evaluate the performance of the following methods developed within our framework, and we also conduct comparisons among these methods to demonstrate the advantages of DPP in rule selection.

- *All 50 Rules*: Average score vectors from all 50 rules to rate and select data.
- *Random 10 Rules*: Randomly choose 10 rules and average the score vectors to rate and select data.
- *DPP 10 Rules*: Use DPP to sample 10 rules, and then apply them to rate and select data.

Note that for uniform sampling and the methods involving randomness in the selection of rules, we considered 3 independent trials and averaged the results (see details in Appendix A.7.3).

## 5.2 GENERAL CONTINUED PRE-TRAINING

We selected 20K samples from our data source using the methods described above for continued pre-training of Pythia-1B, then benchmarked the model's performance. The choice of 20K samples was constrained by our GPU resources. Despite 20K being significantly smaller than the pre-training corpus size, we still observed improvements in benchmark results shown in Table 1, with DPP leading in most metrics. We anticipate these differences will be more obvious in domain-specific fine-tuning settings, demonstrated in 5.3 below.

| Method | ARC-Easy | ARC-Challenge | Winogrande | MMLU | SST-2 | Average |
|---|---|---|---|---|---|---|
| Pythia-1B | 59.8 | 25.2 | 53.5 | 25.6 | 49.0 | 42.6 |
| Uniform Sampling | 59.4 | 24.7 | 53.6 | 25.6 | 49.3 | 42.5 |
| No Rule | 59.6 | 25.1 | 53.7 | 25.6 | 49.2 | 42.6 |
| DSIR | **60.5** | 25.3 | 53.3 | 25.7 | 49.9 | 42.9 |
| QuRating Rules | 59.8 | 25.2 | **54.2** | 25.8 | 49.4 | 42.8 |
| GPT-Uncorrelated | 59.6 | 24.9 | 53.2 | 25.8 | 49.4 | 42.6 |
| All 50 Rules | 60 | 25.3 | 53 | 26.1 | 49.3 | 42.7 |
| Random 10 Rules | 60 | 25.1 | 53.7 | 25.8 | 49.5 | 42.8 |
| DPP 10 Rules | 60 | **25.7** | 54 | **26.2** | **50.1** | **43.2** |

Table 1: General continued pre-training of Pythia-1B using 20K selected data samples from SlimPajama (bold text denotes the first place and underlined text denotes the second place). The first row shows the original model's performance without further training.

## 5.3 DOMAIN-SPECIFC FINE-TUNING

We now focus on domain-specific fine-tuning across four domains: IMDB, Medical, Math, and Code. By selecting 20K domain-related data samples from our source for model training, we aim to enhance domain-specific task performance. As demonstrated in Tables 2 and 3, domain-specific fine-tuning yields more significant improvements than general continued pre-training, where the latter often needs larger datasets to enhance performance due to the broader nature of the training data.

Notably, rule-based methods consistently outperform rule-free approaches in general, especially when comparing against *Random Select* and *No Rule*. Among all rule-based methods, *QuRating* underperforms. As previously noted, such human-designed rules are inherently limited due to varying preferences of designers, introducing bias in rules. Additionally, human designers may not capture the data-dependent correlation between rules effectively. The *GPT-Uncorrelated* rules face a similar issue where the rule selection process is entirely independent of the data. In contrast, our framework begins by automatically generating a diverse set of rules and then selecting an orthogonal subset. Furthermore, our method employs the data-dependent score vector to represent rules and utilizes a quantitative measure to accurately assess their correlation.

Within our framework, DPP demonstrated superior performance compared to using all 50 rules or selecting 10 rules randomly. This aligns with our argument of the importance of rule orthogonality, as well as the intuition that the optimal $r$ is not near the boundaries (both validated by the previous experiments on Section 4). This underscores the effectiveness of a rule-based strategy, which introduces more *balanced* diversity in the data rating aspects and selects better training data. Furthermore, it also demonstrates that our application of DPP in rule selection effectively identifies a core set of high-quality rules, thereby enhancing data quality and ultimately improving model performance.

| Method | IMDB | Medical | | | |
|---|---|---|---|---|---|
| | SA accuracy | college medicine | professional medicine | medical genetics | Medical Average |
| Pythia-1B | 44.5 | 21.4 | 34.2 | 23.0 | 26.2 |
| Uniform Sampling | 43.9 | 23.0 | 42.1 | 22.5 | 28.9 |
| No Rule | 51.1 | 23.1 | 42.6 | 22.0 | 29.2 |
| DSIR | 50.2 | 22.5 | 32.4 | 17.0 | 23.9 |
| QuRating | 47.7 | 21.3 | 42.2 | 22 | 28.5 |
| GPT-Uncorrelated | 50.9 | 23.7 | 42 | 22.7 | 29.4 |
| All 50 Rules | 51 | 23.1 | 42.6 | 23 | 29.6 |
| Random 10 Rules | 51.7 | 24 | 41.2 | 23.5 | 29.6 |
| DPP 10 Rules | **53.5** | **24.6** | **43.3** | **26.8** | **31.6** |

Table 2: IMDB & Medical fine-tuning on Pythia-1B, each using 20K selected data samples from SlimPajama. The first row shows the original model's performance without further training.

| Method | Math | | | | Code | | | | |
|---|---|---|---|---|---|---|---|---|---|
| | elementary | high school | college | Math Average | humaneval | mbpp | multiple-py | multiple-cpp | Code Average |
| Llama3-8B | 41 | 39.6 | 34 | 38.2 | 46.3 | 42.9 | 44 | 48.4 | 45.4 |
| Uniform Sampling | 40.5 | 39.2 | 35 | 38.2 | 38.7 | 38.2 | 38.2 | 39.7 | 38.7 |
| No Rule | 42.3 | 37.4 | 37 | 38.9 | 45.1 | 43.9 | 42.8 | 52.1 | 45.9 |
| DSIR | 41.5 | **41.1** | 34 | 38.9 | 45.1 | 43.6 | **49.1** | 52.2 | 47.5 |
| QuRating | 41.5 | 38.1 | 35 | 38.2 | 43.2 | 43.4 | 40.5 | 45.6 | 43.1 |
| GPT-Uncorrelated | 41.4 | 39 | 37.3 | 39.2 | 41.2 | 43.5 | 39.6 | 48.6 | 43.2 |
| All 50 Rules | 41.8 | 40.7 | 33 | 38.5 | 43.9 | 43.4 | 46.6 | 49.1 | 45.8 |
| Random 10 Rules | 42.9 | 39.8 | 35.2 | 39.3 | 48.5 | 41 | 46.6 | 48.1 | 46 |
| DPP 10 Rules | **43.7** | 40.6 | **38** | **40.8** | **50.5** | **44.2** | 46.9 | **52.7** | **48.6** |

Table 3: Math & Code fine-tuning on Llama3-8B, each using 20K selected data samples from SlimPajama. The first row shows the original model's performance without further training.

## 6 CONCLUSION

We have introduced an automated, rule-based framework for selecting high-quality LLM data, utilizing LLMs to generate a diverse set of rules and the DPP method to eliminate redundancy. Our work is the first to introduce an automated rule evaluation metric and we also propose a rule-based selection pipeline that demonstrates substantial generalizability across various settings, effectively overcoming the limitations of human-designed rules and addressing the challenges associated with the lack of robust rule evaluations. We first demonstrated that our approach enhances the accuracy of data ratings using a dataset with given ground truth scores. Then we conduct experiments that train LLMs with selected data and have shown that our method outperforms various other approaches, both in general pre-training and fine-tuning across four domains. The results indicate that our method successfully generates high-quality, diverse rules, and thereby improves quality of selected data, which in turn leads to improved model performance after trained with the chosen data.

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

# A APPENDIX

## A.1 REPRODUCIBILITY STATEMENT

The necessary code to reproduce our experiments is available in the Anonymous Github repository: `https://anonymous.4open.science/r/DataSelection-F118/`. The repository contains the link to download the data, the functions to calculate the rule correlation and mean squared error, the code for stochastic data sampling, and other utility functions. Moreover, for experiments in Section 4, we have provided the prompts and rules both in Appendix A.6.6 and in our code. For experiments in Section 5, we have provided the details of the training model, training hyperparameters, and GPU information in Appendix A.7.1, and the scripts to run the training and benchmarking steps are in the repository. The related prompts and rules are available in both Appendix A.7.11 and in the code. Also, the exact selected rules are described in Appendix A.7.8.

## A.2 ORTHOGONALITY MEASURES

**Volume of parallelepiped.** In our experiments, we also considered another measure of orthogonality, defined as the "volume" of the parallelepiped formed by vectors. This is mathematically described as:

$$\mathbf{Vol}(\bar{\boldsymbol{S}}) \overset{\text{def}}{=} \frac{\sqrt{\det(\bar{\boldsymbol{S}}^\top \bar{\boldsymbol{S}})}}{\Pi_{i=1}^r \|\boldsymbol{v}_i\|}, \tag{5}$$

where $\boldsymbol{v}_i$ are the columns of $\bar{\boldsymbol{S}}$. The determinant of $\bar{\boldsymbol{S}}^\top \bar{\boldsymbol{S}}$ geometrically represents the squared volume of the parallelepiped formed by the columns of $\bar{\boldsymbol{S}}$ (Kulesza et al., 2012). We normalize by the product of the vector norms since both the magnitude of the vectors and their mutual correlation influence the volume: larger norms increase the volume, whereas higher correlation reduces it. Thus, after normalization, the value of **Vol** serves as an indicator of the overall orthogonality among the column vectors of $\bar{\boldsymbol{S}}$. The phenomena under the usage of this measure are similar to the ones under 1. Therefore we only presented results using the rule correlation.

## A.3 DPP SAMPLING

**Intuition by $r = 2$ case.** Here we use the $r = 2$ case to illustrate the intuition behind DPP and explain why it tends to choose items that are relatively uncorrelated. Using the same notation as in 3.1, let $\boldsymbol{K}$ be the kernel matrix and $\mathcal{Y}, Y$ be the ground set and selected subset respectively. When $r = 2$, consider items $A = \{i, j\}$. Then the probability of both items being selected together is given by:

$$
\begin{aligned}
\mathbb{P}(A \subseteq Y) &= K_{i,i} K_{j,j} - K_{i,j} K_{j,i} \\
&= \mathbb{P}(i \in Y)\mathbb{P}(j \in Y) - K_{i,j}^2 \\
&= \mathbb{P}(i \text{ is chosen})\mathbb{P}(j \text{ is chosen}) - (\text{similarity of items } i, j)^2,
\end{aligned}
$$

since $\boldsymbol{K}$ is symmetric by our definition. Larger similarity of $i, j$ reduces the probability $\mathbb{P}(A \subseteq Y)$, indicating that similar items are less likely to be chosen simultaneously. This underscores the DPP's capacity to promote diversity by favoring the selection of dissimilar items.

**DPP Sampling Algorithm:** The sampling algorithm can be found in Algorithm 1 of Kulesza et al. (2012). The sampling process starts by decomposing the kernel matrix $\boldsymbol{K}$ and involves two main

stages: 1. Selecting eigenvectors by sampling from a Bernoulli distribution based on the eigenvalues, and 2. Sampling a subset from the ground set using an iterative conditional distribution method to ensure diversity, as detailed in (Kulesza et al., 2012). We utilize the `DPPy` Python library (Gautier et al., 2019) for efficient DPP initialization and sampling.

**Time Complexity:** Finding the submatrix (subset of columns) of a matrix to maximize the orthogonality is NP-hard (Civril & Magdon-Ismail, 2007; Kulesza et al., 2012). DPP provides us a relatively good solution. In practice, the computational complexity of sampling from a DPP depends primarily on the eigendecomposition of the kernel matrix $K$. In our case, $K \in \mathbb{R}^{R \times R}$ and therefore it requires $O(R^3)$ time, where $R$ is the number of rules. In the `DPPy` package (Gautier et al., 2019) it uses the spectral sampler by default, so the actual run-time of our DPP implementation is $O(R^3)$.

**DPP Sampling for Data Selection:** We noticed that in a concurrent work Yang et al. (2024), the authors also use DPP to perform data selection, but directly applied to the data itself. However, the approach to directly perform data selection using DPP requires the computation based on the kernel matrix with dimension $N$ (number of samples), which is usually huge in the context of LLM data. Moreover, while DPP inherently prioritizes diversity in data selection, it does not address other quality dimensions. In contrast, our rule-based approach assesses multiple aspects of data quality, ensuring a more comprehensive and robust selection process.

### A.4   STOCHASTIC DATA SELECTION: GUMBEL TOP-$k$ TRICK:

Imagine the cases where the target dataset distribution shows a long-tail pattern with respect to our quality measure, using a deterministic quality score as the cutoff could exclude many possibly valuable data (Albalak et al., 2024). Hence, our stochastic sampling in 3 effectively balances the quality and diversity of the selected data. Nonetheless, instead of doing actual sampling according to Equation 3, we use the Gumbel top-$k$ trick similar as in (Wettig et al., 2024), which is a sampling technique used to efficiently and probabilistically select the top-$k$ items from a discrete probability distribution. Specifically, each item $i$ in the distribution is assigned a score using the formula:

$$s_i = \log p_i + g_i,$$

where $p_i$ is the probability of item $i$, and $g_i$ is a noise term drawn from a Gumbel distribution, which can be generated using $g_i = -\log(-\log(u_i))$. In other words, we could add a Gumbel noise vector to the log of the sampling probability in Equation 3 and then choose the top-$k$ data points with the highest sums. This is statistically equivalent to sampling according to Equation 3 (Kool et al., 2019).

### A.5   LIMITATIONS AND FUTURE DIRECTIONS

We have developed an automated, rule-based selection framework for identifying high-quality LLM data. Below, we outline some limitations of our approach and suggest potential directions for future research:

**Adjusting hyperparameters.** Recall that our hyperparameter $r$ determines the number of rules selected for rating, influencing the diversity and coverage of the selected rules. We have explored the effect of $r$ in Section 4 and also in Appendix A.7.5. We leave a comprehensive study of its optimal values for future work.

**Data sampling method.** There are variations of the stochastic top-$k$ sampling, such as incorporating a temperature parameter $\tau$ (see Wettig et al. (2024)). Replacing equation 3 with its variations or exploring other data sampling methods represents another research direction.

**Rule format.** In this study, we only focus on natural language rules, which are straightforward to design and offer significant explainability. However, rules in other formats can also be integrated into our pipeline.

**Other rule evaluations metrics.** We have proposed multiple metrics in 1 and A.2 to measure rule quality, but all based on the correlation/orthogonality of rules. Evaluating rules from other aspects is another intriguing topic for future work.

### A.6 Appendix for Evaluation A

#### A.6.1 Bradley Terry Model

The Bradley-Terry model is a probabilistic model used to estimate the latent "strength" of teams based on pairwise competitions. The model is parameterized as follows:

$$\mathbb{P}(i \text{ beats } j) = \frac{v_i}{v_i + v_j} = \frac{e^{\beta_i}}{e^{\beta_i} + e^{\beta_j}}, \tag{6}$$

where exponential functions are used to model the scores $v_i \overset{\text{def}}{=} e^{\beta_i}$ and $v_j \overset{\text{def}}{=} e^{\beta_j}$. In other words, the difference of their scores determines the the log-odds of team $i$ beating team $j$. Sometimes an intercept term $\alpha$ is added to adjust for any influence of the order (for example, imagine that $i$ is the home team and has home-court advantage), then the probability becomes

$$\mathbb{P}(i \text{ beats } j) = \frac{e^{\alpha+\beta_i}}{e^{\alpha+\beta_i} + e^{\beta_j}}, \tag{7}$$

The most straightforward method for estimating these parameters is through maximum likelihood estimation, which optimizes the likelihood of the observed outcomes based on the model and its parameters. More details can be found in Bradley & Terry (1952); Hunter (2004).

#### A.6.2 Error Metrics

**Ranking-difference error.** To assess the deviation of rating scores from the ground truth, instead of using the mean squared error in 4, an alternative intuitive approach is to compare the rankings derived from the data scores with those of the ground truth. This approach is based on the premise that for data selection purposes, if two sets of scores yield identical rankings, they will select the same high-scoring data samples. An example of such a ranking metric is the Kendall rank correlation coefficient (Kendall's tau) (Kendall, 1938). However, we opted against this type of metric for two critical reasons: First, it lacks the granularity needed to evaluate errors effectively. For instance, two sets of scores like [0.01, 0.98, 0.99] and [0.01, 0.02, 0.03] share exactly the same ranking yet differ significantly in their actual scores. Second, our method involves stochastic data selection, not a straightforward top-$k$ selection, meaning that a higher score increases the likelihood of a data point being chosen. Hence, a ranking difference, which overlooks the absolute values of scores and focuses solely on their relative comparisons, is not ideal here.

#### A.6.3 Rating scheme Variation: Individual rating

Here we present the results after replacing the pair-wise rating with the direct individual rating in Section 4:

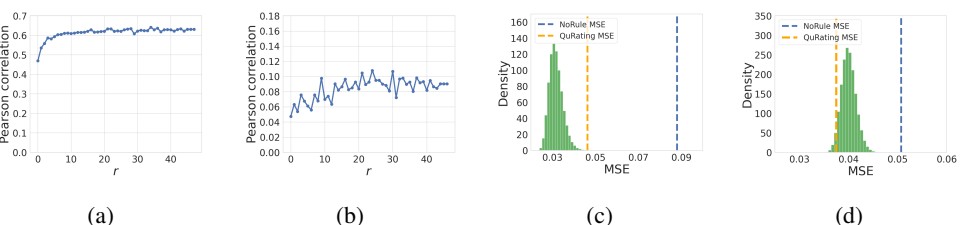

(a)        (b)        (c)        (d)

Figure 4: (a) and (b): Pearson correlation of the rule correlation $\rho(\bar{S})$ and the MSE $\epsilon(\bar{S})$, for IMDB and CommonCrawl datasets respectively. (c) and (d): Distribution of MSE from $10^6$ possible rule subsets with size $r$, for IMDB and CommonCrawl datasets respectively. Two vertical lines represent the MSE values of QuRating and NoRule.

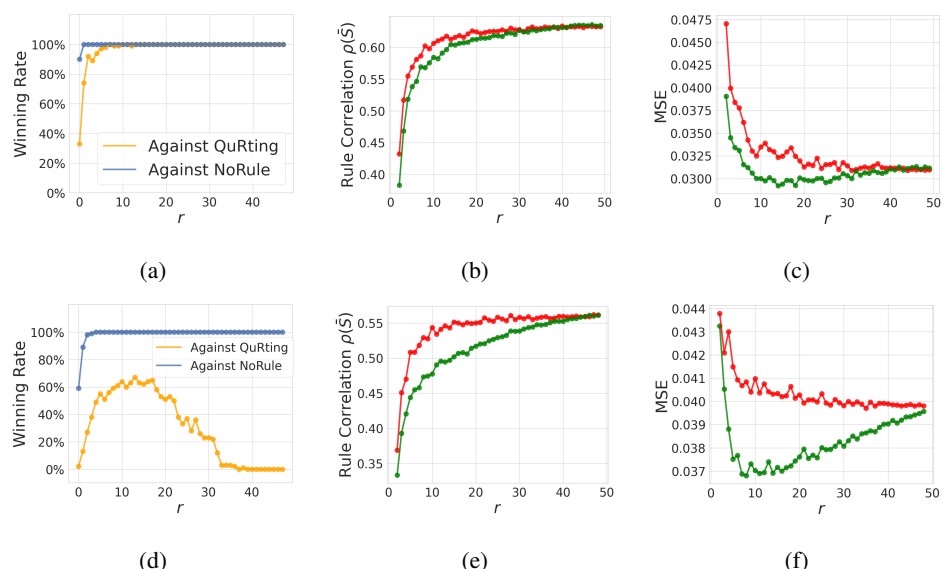

(a)                              (b)                              (c)

(d)                              (e)                              (f)

Figure 5: (a) Winning rate of DPP-selected rules compared to QuRating's four rules and the NoRule setting, based on MSE across 100 DPP trials. (b) Comparison of DPP rule correlation vs. random rule correlation (averaged over 100 trials). (c) Comparison of MSE between DPP-selected and randomly selected rules, averaged across 100 trials. Plots (a), (b), and (c) display results for the IMDB dataset, while (d), (e), and (f) for the CommonCrawl dataset.

### A.6.4   RATER MODEL SIZE VARIATION: LLAMA3-70B-INSTRUCT

Here we present the results after replacing the rater model from Llama3-8B-Instruct model with the stronger Llama3-70B-Instruct in Section 4:

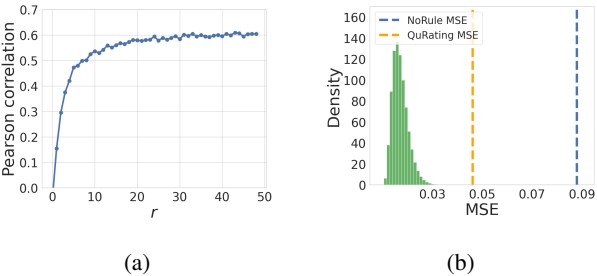

(a)                              (b)

Figure 6: (a): Pearson correlation of the rule correlation $\rho(\bar{S})$ and the MSE $\epsilon(\bar{S})$ (b): Distribution of MSE from $10^6$ possible rule subsets with size $r$. Two vertical lines represent the MSE values of QuRating and NoRule.

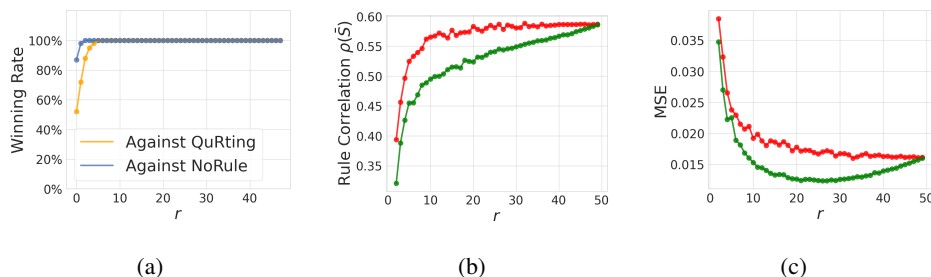

(a)            (b)            (c)

Figure 7: (a) Winning rate of DPP-selected rules compared to QuRating's four rules and the NoRule setting, based on MSE across 100 DPP trials. (b) Comparison of rule correlation between DPP-selected and randomly selected rules, averaged across 100 trials. (c) Comparison of MSE between DPP-selected and randomly selected rules, averaged across 100 trials.

### A.6.5 RULE GENERATOR VARIATION: CLAUDE-3.5-SONNET

To verify that GPT-4 is a reliable rule generator, we compare it with Claude-3.5-Sonnet. For each of the five tasks (General, IMDB, Medical, Math, Code), we prompt GPT and Claude to generate 100 rules for each, and then study the distribution of the rules. Specifically, we use Sentence-Transformer (Reimers, 2019) to generate the embedding vectors and then use PCA to project them onto the top two principal components for 2-dimensional visualization. From Figure 8 below, we observe that the two groups of rules generated by the two models completely overlap, demonstrating no distinct separation. This suggests that GPT-4 functions effectively as an unbiased rule generator.

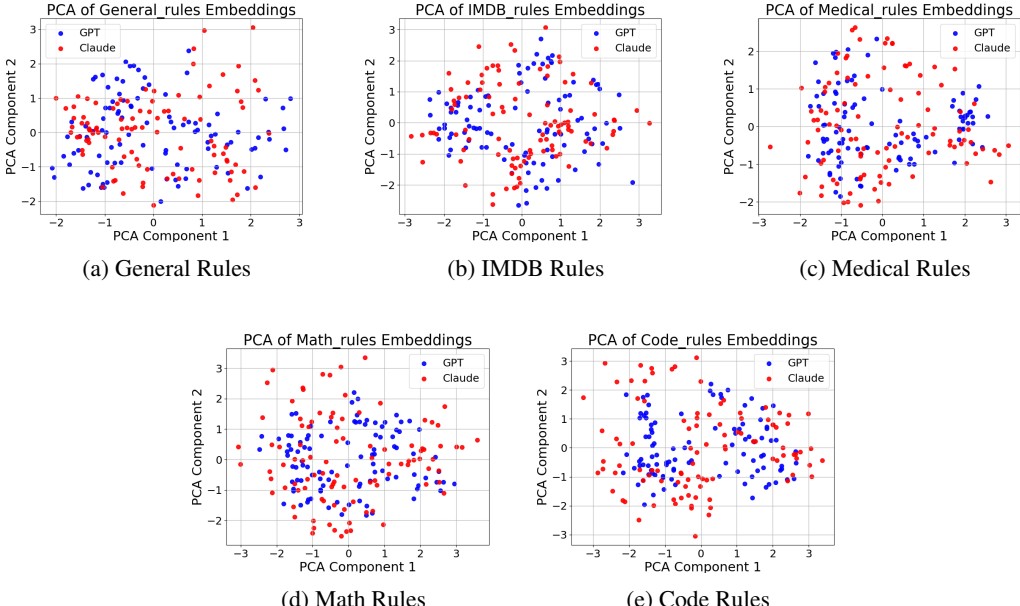

(a) General Rules      (b) IMDB Rules      (c) Medical Rules

(d) Math Rules      (e) Code Rules

Figure 8: Embeddings of rules generated by GPT and Claude across different domains.

To further quantify the distribution differences, we studied the Wasserstein distance within and between the two rule sets. Specifically, we compute the distance between the GPT rules and Claude rules. Then we randomly split GPT rules into two parts and computed their Wasserstein distance and similarly for the Claude rules (averaged over 10 trials). By comparing these values (see Table A.6.5), we found no clear distribution bias when switching from one rule generator to the other.

Table 4: Comparison of Intra-Model and Inter-Model Metrics across Domains

|             | General | IMDB  | Medical | Math  | Code  |
|-------------|---------|-------|---------|-------|-------|
| Intra-GPT   | 0.432   | 0.438 | 0.502   | 0.584 | 0.673 |
| Intra-Claude| 0.455   | 0.445 | 0.515   | 0.592 | 0.668 |
| Inter-Model | 0.468   | 0.548 | 0.522   | 0.612 | 0.684 |

Now we have compared the rule generators using GPT and Claude above. In order to address potential biases in rule generation by these large language models (LLMs) compared to human-generated rules, we prompt GPT-4 to generate 133 ethical and safety rules and compare the GPT-generated rules with the public and standard constitutions in Huang et al. (2024) (we make the rules all start from "Choose the response that" for a fair comparison). We asked 3 authors who have not seen the constitutions in Huang et al. (2024) to distinguish the rules blindly. We get an average accuracy of 20.7%, suggesting it is indeed hard to distinguish between rules generated by GPT and those designed by humans. All these discussions underscores the potential of GPT as a reliable rule generator that is capable of producing rules that are comparable to those crafted by human experts.

### A.6.6 PROMPTS AND GENERATED RULES

**Comparison prompt:** Below is the template used to compare two data samples according to a specific rule. For the rule-free version, simply omit the sentence involving the rule. Replace DATASET_NAME with "IMDB reviews" or "Common Crawl data" to correspond to the two data sources discussed in Section 4.

Compare two data examples from `<DATASET_NAME>` and choose the example which has better quality according to the following rule:
`<RULE>`

The texts might have similar quality, but you should still make a relative judgement and choose the label of the preferred text.
Example A:
`<DATA_SAMPLE A>`

Example B:
`<DATA_SAMPLE B>`

Now you have to choose between either A or B. You must respond only with a single letter 'A' or 'B'.

Figure 9: Template of rule-based comparison prompt.

**Generated IMDB rules:**

| Index | Rule and Description |
|-------|----------------------|
| 0 | Clearly state the main opinion or sentiment of the reviewer. |
| 1 | Be free of spelling errors. |
| 2 | Be free of grammatical errors. |
| 3 | Have a coherent structure with a clear beginning, middle, and end. |
| 4 | Be relevant to the movie being reviewed. |
| 5 | Avoid using offensive or inappropriate language. |
| 6 | Provide specific reasons for the given sentiment. |
| 7 | Include details that support the overall sentiment. |
| 8 | Be free from excessive use of exclamation marks. |
| 9 | Not contain any personal attacks on individuals. |
| 10 | Not be overly repetitive. |

| Index | Rule and Description |
|-------|---------------------|
| 11 | Avoid vague statements and provide concrete examples. |
| 12 | Not include spoilers without a spoiler warning. |
| 13 | Be written in complete sentences. |
| 14 | Not use excessive capitalization for emphasis. |
| 15 | Have a logical flow and avoid jumping between unrelated points. |
| 16 | Not contain any irrelevant information. |
| 17 | Be at least 100 words long. |
| 18 | Not exceed 500 words. |
| 19 | Not contain any text that is irrelevant to the movie review. |
| 20 | Not include any links or advertisements. |
| 21 | Provide a balanced perspective, mentioning both positives and negatives if applicable. |
| 22 | Not be biased or prejudiced. |
| 23 | Be written from a first-person perspective. |
| 24 | Not contain any misleading information. |
| 25 | Mention the movie title at least once. |
| 26 | Be engaging and hold the reader's attention. |
| 27 | Avoid overly technical language that might confuse readers. |
| 28 | Not be a duplicate of another review in the dataset. |
| 29 | Mention specific scenes or elements of the movie when providing critiques. |
| 30 | Provide a final summary of the reviewer's overall opinion. |
| 31 | Not include excessive punctuation marks such as multiple question marks or exclamation points. |
| 32 | Not use text abbreviations or slang. |
| 33 | Be written in a formal or semi-formal tone. |
| 34 | Provide context for any cultural or historical references. |
| 35 | Not make unsupported generalizations. |
| 36 | Maintain a consistent tone throughout. |
| 37 | Not contradict itself. |
| 38 | Indicate whether the reviewer recommends the movie or not. |
| 39 | Not contain any unnecessary filler words or phrases. |
| 40 | Be respectful and considerate in its critique. |
| 41 | Address the acting, direction, and cinematography if possible. |
| 42 | Be free from any copy-pasted text from other sources. |
| 43 | Not include any personal anecdotes unrelated to the movie. |
| 44 | Be specific about what worked and what didn't in the movie. |
| 45 | Mention the genre of the movie. |
| 46 | Include a rating or score if available. |
| 47 | Be written with the target audience in mind. |
| 48 | Provide insights into the movie's themes and messages. |
| 49 | Not contain any text that is purely promotional in nature. |

Table 5: Generated 50 Rules for rating IMDB examples.

**Generated CommonCrawl rules:**

| Index | Rule Description |
|-------|------------------|
| 0 | The text should be free of spelling errors. |
| 1 | The grammar should be correct and appropriate for the context. |
| 2 | The content should be relevant to the topic described in the title or metadata. |
| 3 | The text should not contain any offensive or inappropriate language. |
| 4 | The information presented should be factually accurate. |
| 5 | The text should not be overly repetitive. |
| 6 | The sentences should be clear and concise. |
| 7 | The text should provide useful and meaningful information. |
| 8 | The content should be engaging and interesting to the reader. |
| 9 | The text should have a logical flow and coherent structure. |
| 10 | The text should not contain broken or incomplete sentences. |
| 11 | The metadata should accurately reflect the content of the text. |
| 12 | The text should not include excessive jargon or overly complex language. |
| 13 | The content should be relevant to the intended audience. |

| Index | Rule Description |
|---|---|
| 14 | The text should be free of any advertisements or promotional material. |
| 15 | The text should not contain any personal information or sensitive data. |
| 16 | The content should be original and not plagiarized. |
| 17 | The text should not include any irrelevant or off-topic information. |
| 18 | The text should be formatted properly with appropriate headings and paragraphs. |
| 19 | The text should be free of any links or URLs unless relevant and necessary. |
| 20 | The content should be up-to-date and not outdated. |
| 21 | The text should be free of any empty or meaningless filler words. |
| 22 | The content should provide a balanced and unbiased perspective. |
| 23 | The text should not contain any images or multimedia unless relevant and properly embedded. |
| 24 | The text should have proper punctuation marks. |
| 25 | The text should not contain any placeholders or unfinished sentences. |
| 26 | The text should be suitable for the language model's training purposes. |
| 27 | The text should maintain a consistent tone and style throughout. |
| 28 | The text should not contain any HTML or other markup language unless specified. |
| 29 | The text should avoid slang or colloquial expressions unless contextually appropriate. |
| 30 | The content should have proper citations or references if necessary. |
| 31 | The text should be sufficiently detailed to provide value to the reader. |
| 32 | The text should be free of any biased or prejudiced language. |
| 33 | The text should not contain any technical errors or glitches. |
| 34 | The content should have a clear beginning, middle, and end. |
| 35 | The text should be free of any redundant phrases or statements. |
| 36 | The text should adhere to any specified length requirements. |
| 37 | The text should not contain any duplicate content. |
| 38 | The text should be relevant to the specified geographic location if mentioned. |
| 39 | The text should not include any speculative or unverified information. |
| 40 | The content should encourage reader engagement and interaction. |
| 41 | The text should maintain a professional tone unless otherwise specified. |
| 42 | The text should be free of any ambiguities or unclear statements. |
| 43 | The content should not promote any illegal activities or behaviors. |
| 44 | The text should have a neutral point of view unless otherwise specified. |
| 45 | The text should be free of any distracting formatting errors. |
| 46 | The content should address any specified keywords or topics effectively. |
| 47 | The text should be free of any content that violates copyright or intellectual property rights. |
| 48 | The text should have a clear and relevant title or headline. |
| 49 | The text should be suitable for training language models for general downstream tasks. |

Table 6: Generated 50 Rules for rating CommonCrawl examples.

## A.7 APPENDIX FOR EVALUATION B

### A.7.1 MODEL TRAINING

For training Pythia-1B and Llama3-8B, we loaded both models using `bfloat16` precision and used one `NVIDIA A100-80GB` for each training job. Below are the training parameters:

Table 7: Comparison of Model Parameters

| Model | Pythia-1B | Llama3-8B |
|---|---|---|
| **Num of epochs** | 1 | 1 |
| **Batch size** | 1 | 1 |
| **Learning rate** | $2 \cdot 10^{-5}$ | $2 \cdot 10^{-5}$ |
| **Token max length** | 2048 | 4096 |
| **LoRA** | No | Yes (rank=64) |

### A.7.2 MORE BASELINE METHODS

Here we add two more baseline methods: *LESS* (Xia et al., 2024): selecting data based on the estimated data influences, and *DiverseEvol* (Wu et al., 2023): an iterative sampling algorithm to ensure

data diversity. It is important to note that *DiverseEvol* focuses solely on a single quality aspect: the diversity of data, while our method ensures diversity across multiple rating aspects. Another remark is that in the original papers, these methods were specifically used for instruction tuning data. We copy the three rule-based methods in our framework from 1 for comparison purposes. From the results below, we see these two methods, while being computationally expensive, are not showing good performance under our experiment settings.

| Method | ARC-Easy | ARC-Challenge | Winogrande | MMLU | SST-2 | Average |
|---|---|---|---|---|---|---|
| LESS | 59.8 | 25.2 | 52.9 | 25.6 | 49.3 | 42.5 |
| DiverseEvol | 59.7 | 24.7 | 53.3 | 25.6 | 49.2 | 42.5 |
| All 50 Rules | 60 | 25.3 | 53 | 26.1 | 49.3 | 42.7 |
| Random 10 Rules | 60 | 25.1 | 53.7 | 25.8 | 49.5 | 42.8 |
| DPP 10 Rules | 60 | **25.7** | 54 | **26.2** | **50.1** | **43.2** |

Table 8: General continued pre-training of Pythia-1B using 20K selected data samples from SlimPajama.

| Method | IMDB | Medical | | | |
|---|---|---|---|---|---|
| | SA accuracy | college medicine | professional medicine | medical genetics | Medical Average |
| LESS | 46.6 | 23.6 | 40.4 | 24 | 29.3 |
| DiverseEvol | 51.1 | 23.6 | 42.5 | 23 | 29.7 |
| All 50 Rules | 51 | 23.1 | 42.6 | 23 | 29.6 |
| Random 10 Rules | 51.7 | 24 | 41.2 | 23.5 | 29.6 |
| DPP 10 Rules | **53.5** | **24.6** | **43.3** | **26.8** | **31.6** |

Table 9: IMDB & Medical fine-tuning on Pythia-1B, each using 20K selected data samples from SlimPajama.

| Method | Math | | | | Code | | | | |
|---|---|---|---|---|---|---|---|---|---|
| | elementary | high school | college | Math Average | humaneval | mbpp | multiple-py | multiple-cpp | Code Average |
| LESS | 41.5 | 40.4 | 33 | 38.3 | 41.4 | 43.5 | 43.9 | 45.3 | 43.5 |
| DiverseEvol | 41.2 | 38.5 | 35 | 38.2 | 38.4 | 43.6 | 42.8 | 47.8 | 43.1 |
| All 50 Rules | 41.8 | 40.7 | 33 | 38.5 | 43.9 | 43.4 | 46.6 | 49.1 | 45.8 |
| Random 10 Rules | 42.9 | 39.8 | 35.2 | 39.3 | 48.5 | 41 | 46.6 | 48.1 | 46 |
| DPP 10 Rules | **43.7** | 40.6 | **38** | **40.8** | **50.5** | **44.2** | 46.9 | **52.7** | **48.6** |

Table 10: Math & Code fine-tuning on Llama3-8B, each using 20K selected data samples from SlimPajama.

### A.7.3 VARIANCE OF TRIALS

Due to computational resource constraints, we were unable to perform multiple repetitions of all experiments. However, as mentioned in Section 5, we conducted 3 independent trials in four domains for *Uniform Sampling* and methods involving randomness in rule selections, including *GPT-Uncorrelated*, *Random 10 Rules*, and *DPP 10 Rules* (note that DPP sampling is also non-deterministic) to mitigate the effects of randomness, and we report their standard deviations here.

| Method | IMDB | Medical | | | |
|---|---|---|---|---|---|
| | SA accuracy | college medicine | professional medicine | medical genetics | Medical Average |
| Uniform Sampling | $43.9_{1.1}$ | $23.0_{0.42}$ | $42.1_{0.62}$ | $22.5_{0.71}$ | $28.9_{0.77}$ |
| GPT-Uncorrelated | $50.9_{0.71}$ | $23.7_{0.11}$ | $42_{0.3}$ | $22.7_{0.76}$ | $29.4_{0.2}$ |
| Random 10 Rules | $51.7_{0.21}$ | $24_{0.42}$ | $41.2_{1.2}$ | $23.5_{0.41}$ | $29.6_{0.58}$ |
| DPP 10 Rules | $53.5_{0.58}$ | $24.6_{0.37}$ | $43.3_{0.43}$ | $26.8_{0.76}$ | $31.6_{0.41}$ |

Table 11: Mean and standard deviation over 3 independent trials for the IMDB & Medical fine-tuning setting.

| Method | Math | | | | Code | | | | |
|---|---|---|---|---|---|---|---|---|---|
| | elementary | high school | college | Math Average | humaneval | mbpp | multiple-py | multiple-cpp | Code Average |
| Uniform Sampling | $40.5_{0.35}$ | $39.2_{0.56}$ | $35_{0.2}$ | $38.2_{0.30}$ | $38.7_{1.27}$ | $38.2_{1.6}$ | $38.2_{0.42}$ | $39.7_{1.2}$ | $38.7_{0.94}$ |
| GPT-Uncorrelated | $41.4_{0.17}$ | $39_{0.21}$ | $37.3_{0.57}$ | $39.2_{0.28}$ | $41.2_{0.15}$ | $43.5_{0.11}$ | $39.6_{1.44}$ | $48.6_{0.15}$ | $43.2_{0.32}$ |
| Random 10 Rules | $42.9_{0.1}$ | $39.8_{1.3}$ | $35.2_{0.28}$ | $39.3_{0.48}$ | $48.5_{0.6}$ | $41_{0.94}$ | $46.6_{0.85}$ | $48.1_{1.2}$ | $46_{0.78}$ |
| DPP 10 Rules | $43.7_{0.61}$ | $40.6_{0.32}$ | $38_{0}$ | $40.8_{0.15}$ | $50.5_{0.36}$ | $44.2_{0.26}$ | $46.9_{0.2}$ | $52.7_{0.15}$ | $48.6_{0.22}$ |

Table 12: Mean and standard deviation over 3 independent trials for the Math & Code fine-tuning setting.

Here we perform the $t$-test to demonstrate that the advantage of *DPP 10 Rules* is significant compared to other methods. We include the $t$-statistics and $p$-values in the table. If we choose the significance threshold $p = 0.05$, then we see that all the comparisons are significant.

| Comparison | IMDB | Medical average |
|---|---|---|
| DPP vs GPT-Uncorrelated | $t = 4.912, p = 0.00881$ | $t = 8.353, p = 0.00408$ |
| DPP vs Uniform Sampling | $t = 13.371, p = 0.00086$ | $t = 5.361, p = 0.01218$ |
| DPP vs Random 10 Rules | $t = 5.054, p = 0.02255$ | $t = 4.877, p = 0.01065$ |

Table 13: Comparison of DPP method with other methods in IMDB and Medical AVG domains using Welch's t-test.

| Comparison | Math average | Code average |
|---|---|---|
| DPP vs GPT-Uncorrelated | $t = 8.724, p = 0.00293$ | $t = 24.085, p = 0.00005$ |
| DPP vs Uniform Sampling | $t = 13.426, p = 0.00099$ | $t = 17.762, p = 0.00195$ |
| DPP vs Random 10 Rules | $t = 5.166, p = 0.02415$ | $t = 5.557, p = 0.02204$ |

Table 14: Comparison of DPP method with other methods in Math and Code domains using Welch's t-test.

### A.7.4 EVALUATION BENCHMARKS

In this section, we provide detailed descriptions of the benchmarks utilized for our evaluation. We considered the following benchmarks for general continued pre-training: ARC-Challenge (15), Winogrande (15), MMLU (5), SST-2 (0), where the numbers in parenthesis indicate the number of shots we use in few-shot benchmark setting. For domain fine-tuning, we use zero-shot in IMDB, and 5-shot for Medical and Math (which uses subsets of MMLU). Moreover, Math and Medical domains, we use the subject-related subsets from MMLU, specifically ElementaryMathematics, HighSchoolMathematics, and CollegeMathematics for Math, and CollegeMedicine, ProfessionalMedicine, and MedicalGenetics for Medical. For Code, we tested code generation and for each code benchmark, we use the pass@k setting and specify the number of code generation samples. See detailed explanations below.

- **MMLU** (Maas et al., 2011): MMLU is a comprehensive multitask test comprises multiple-choice questions from a wide range of knowledge domains. It spans subjects across the humanities, social sciences, hard sciences, and other critical learning areas, encompassing 57 tasks such as elementary mathematics, US history, computer science, law, and more. To achieve high accuracy on this test, models need to demonstrate extensive world knowledge and robust problem-solving capabilities.

- **IMDB** (Maas et al., 2011): The IMDB dataset comprises 50,000 movie reviews and is designed for binary sentiment classification. For our evaluation, we select 25,000 test samples.

- **Winogrande** (Sakaguchi et al., 2021): WinoGrande is a collection of 44,000 problems inspired by the Winograd Schema Challenge. It has been adjusted to enhance scale and robustness against dataset-specific bias. Designed as a fill-in-the-blank task with binary options, WinoGrande requires users to select the correct option for a given sentence based on commonsense reasoning.

- **SST-2** (Socher et al., 2013): SST-2, or the Stanford Sentiment Treebank binary classification dataset, is a widely used resource for sentiment analysis tasks. Derived from movie reviews, it consists of 11,855 single sentences, each annotated for sentiment polarity.

- **ARC-Easy and ARC-Challenge** (Yadav et al., 2019): The AI2's Reasoning Challenge (ARC) dataset is designed for evaluating multiple-choice question-answering systems. It consists of science exam questions for grades 3 to 9 and is divided into two subsets: Easy and Challenge. The Challenge subset comprises more complex questions that necessitate advanced reasoning skills. Typically, questions offer four answer choices, although a small fraction (less than 1%) may present three or five options. The dataset also features a Knowledge Base (KB) containing 14.3 million unstructured text passages to support reasoning and answer generation.

- **HumanEval** (Chen et al., 2021): The HumanEval benchmark evaluates Python programming skills with 164 problems, each comprising a function signature, docstring, function body, and unit tests. In a zero-shot setting, models generate code using top-p sampling (p=0.95) until stop words are reached. Pass@k metrics (k=1, 10, 100) are calculated with n=200 samples per problem, estimating the success rate following Chen et al.'s approach. Success is determined by whether at least one solution is correct within k attempts, with temperature controlling randomness in generation. This benchmark measures model performance in solving programming tasks with increasing attempts.

- **MBPP** (Austin et al., 2021): The MBPP benchmark contains around 1,000 crowd-sourced Python programming problems, designed for entry-level programmers. Each problem includes a task description, a code solution, and 3 test cases. The evaluation is performed on the test set from index 11 to 511. In a few-shot setting, the InCoder-style prompt is used, where the task description and one solution are provided to guide the model. The prompt format is `f'"""{description}{test_example}"""'`. By default, `prompt_type_mbpp` is set to `incoder`, and optionally, the solution can be included using `include_solution_mbpp=True`. We use single generation per problem (pass@1), and for pass@k estimation, we generate n=15 samples per problem, similar to the HumanEval approach. The evaluation focuses on pass@1 success rates.

- **Multiple-py and Multiple-cpp** (Cassano et al., 2022): MultiPL-E: is a benchmark for evaluating large language models for code generation that supports 18 programming languages. It takes the OpenAI "HumanEval" Python benchmark and uses little compilers to translate them to other languages. We use similar implementation as the original repository and evaluation parameters are similar to HumanEval.

### A.7.5 NUMBER OF SELECTED RULES

We modified the number of rules, $r$, from 10 to 20 and repeated the experiments for the Code domain. Compared to the 10-rule results presented in Table 3, we observed some discrepancies. For instance, the performance score on HumanEval is less than the 10-rule results, whereas the results for Multiple-cpp improved. The number of rules indeed alters the criteria used for data selection, thereby influencing the distribution of the selected data. Determining the optimal $r$ represents a valuable direction for future exploration.

| Method | Code | | | | |
|---|---|---|---|---|---|
| | humaneval | mbpp | multiple-py | multiple-cpp | Code Average |
| Random 20 Rules | 43.90 | 43.93 | 46.57 | 50.50 | 46.23 |
| DPP 20 Rules | 45.10 | 44.80 | 49.10 | 53.40 | 48.10 |

Table 15: Code fine-tuning on Llama3-8B using 20K selected data samples from our SlimPajama data source. Instead of using 10 rules, 20 rules were selected during the rule selection step.

### A.7.6 SIZE OF SAMPLED DATA

We investigated the impact of varying training data sizes on performance, specifically within the context of the *DPP 10 rules* and the Medical domain. Our observations reveal that increasing the amount of training data does not always enhance performance; in fact, performance may decline

beyond a certain data threshold. This phenomenon is consistent with findings from the LIMA paper (Zhou et al., 2024), which suggests that data quality is often more important than quantity for LLMs. Balancing data quality with quantity is another challenging but valuable topic.

| Training Size | Medical | | | |
|---|---|---|---|---|
| | college medicine | professional medicine | medical genetics | Medical average |
| 10K | 23.1 | 41.5 | 27.0 | 30.5 |
| 20K | 24.3 | 43.0 | 26.0 | 31.1 |
| 50K | 23.7 | 44.5 | 24.0 | 30.7 |
| 100K | 23.7 | 39.3 | 24.0 | 29.0 |
| 200K | 23.1 | 42.6 | 21.0 | 28.9 |

Table 16: Medical fine-tuning on Pythia-1B using various sizes of training data selected by DPP with 10 rules.

### A.7.7 DISTRIBUTION OF SELECTED DATA

Evaluating and contrasting the quality of data subsets selected by different methods is challenging and often necessitates extensive human intervention. To address this, the authors examined the initial 100 examples selected by each method. This examination revealed notable distinctions in the relevance and domain specificity of the data selected. Specifically, our DPP rule-based approach demonstrated a marked ability to identify and select examples that were highly pertinent to specific domains. For instance, in experiments focused on the Code domain, this method favored the inclusion of data containing code. In contrast, other less targeted methods, such as QuRating and Uniform Sampling, often yield selections that lack domain-specific relevance. This insight underscores the efficacy of using tailored, rule-based methods over generic ones for tasks where domain alignment is critical.

Although it is hard to compare the distribution of the selected data, we provide a visual representation in Figure 10 below, showcasing the meta-data (categories of the data samples) distributions for the Code domain as a representative example. Notably, the DPP methods with 10 and 50 rules tend to select more data from GitHub and StackExchange for Code fine-tuning.

Moreover for IMDB domain, in Figure 11 we investigated the text length distribution. We see that the QuRating is very close to the original SlimiPajama distribution, where we conjecture that in this case the data distribution is very close to uniformly sampled data. The methods within our framework have a tendency toward longer texts. Additionally, in Figure 12 we use bigram entropy (the Shannon entropy of the distribution over the unique bigrams) as an indicator of the text diversity. We again see that the entropy distribution of QuRating is very close to the original SlimPajama, where our methods generally select data with higher entropy/diversity and the entropy distributions are more concentrated.

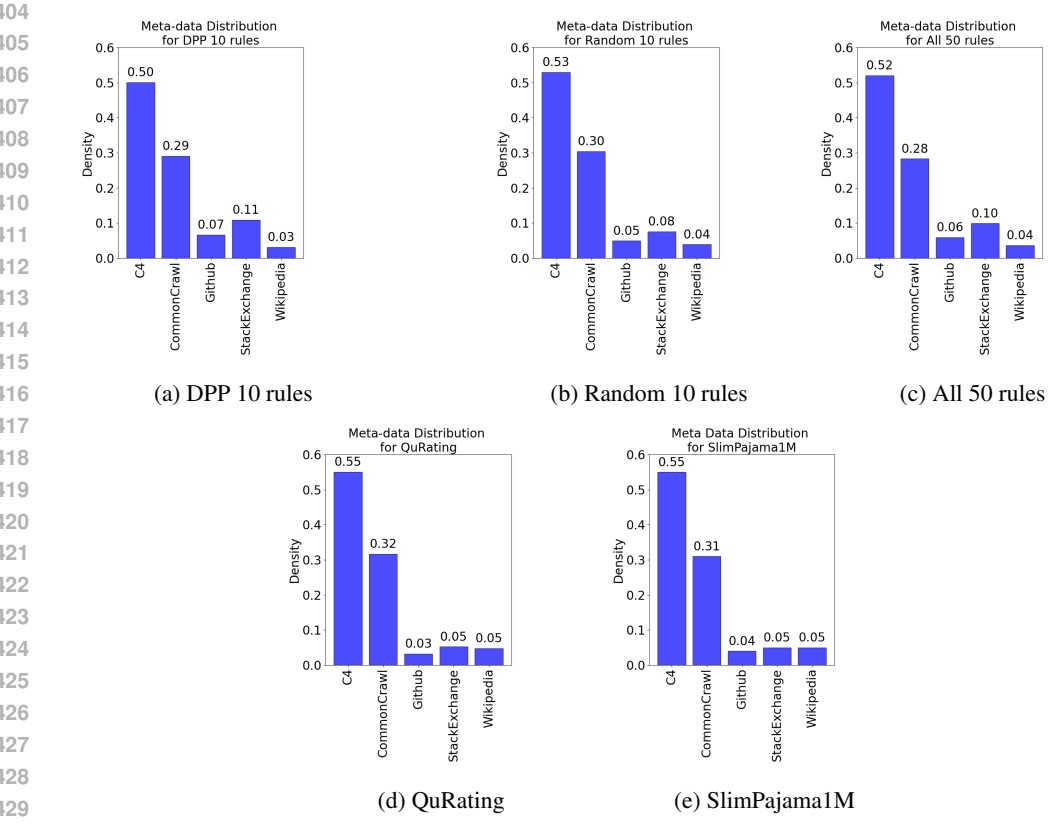

(a) DPP 10 rules      (b) Random 10 rules      (c) All 50 rules

(d) QuRating      (e) SlimPajama1M

Figure 10: Comparison of meta-data distribution across different methods. The last is the original distribution of our source data.

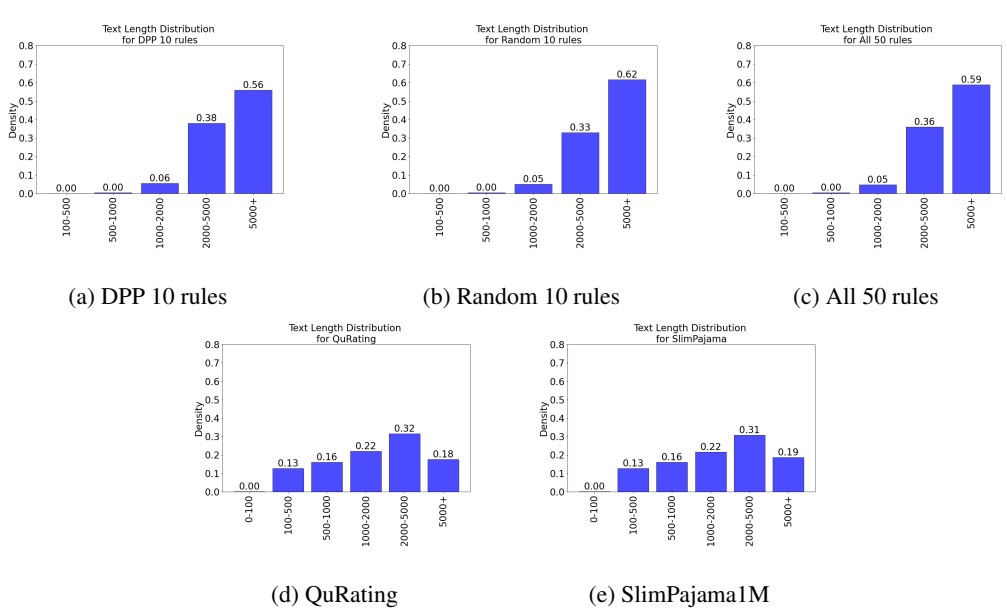

(a) DPP 10 rules      (b) Random 10 rules      (c) All 50 rules

(d) QuRating      (e) SlimPajama1M

Figure 11: Comparison of text length distribution across different methods. The last is the original distribution of our source data.

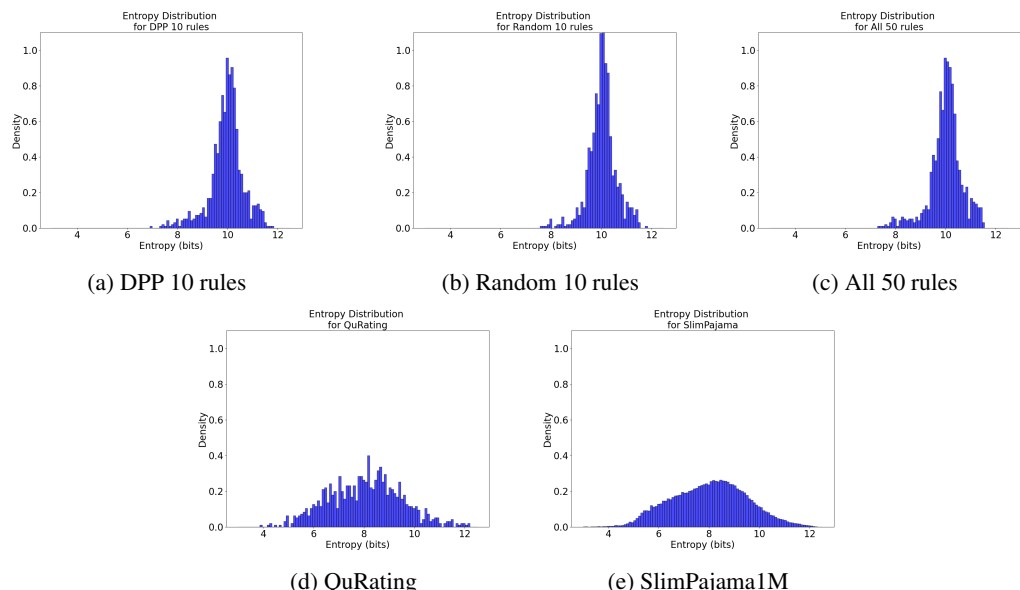

(a) DPP 10 rules    (b) Random 10 rules    (c) All 50 rules

(d) QuRating    (e) SlimPajama1M

Figure 12: Comparison of text diversity distribution across different methods. The last is the original distribution of our source data.

### A.7.8 RULE CORRELATION OF SELECTED RULES

Here we provide in Table 17 the rule indices (in the range $\{0, 1, \ldots, 49\}$) for the rules selected by DPP and random selection (in one trial). The fullest of all generated 50 rules for each domain is provided in A.7.11. For each set of selected rules, we also calculate their rule correlation value $\rho$ (defined in 1). We confirm that indeed DPP selects rules with lower rule correlation than random selected rules.

Table 17: Rule correlation and indices of the selected rules by DPP and random selection.

| Domain | Method | Rule Indices |
|---|---|---|
| IMDB | DPP 10 rules ($\rho = 0.42$) | [2, 3, 13, 21, 28, 36, 37, 45, 46, 49] |
| | Random 10 rules A ($\rho = 0.53$) | [2, 6, 10, 11, 15, 21, 28, 42, 43, 48] |
| | Random 10 rules B ($\rho = 0.51$) | [1, 9, 12, 14, 25, 26, 27, 37, 38, 40] |
| Medical | DPP 10 rules ($\rho = 0.55$) | [1, 9, 10, 25, 29, 30, 32, 38, 42, 47] |
| | Random 10 rules A ($\rho = 0.69$) | [11, 13, 14, 16, 25, 33, 34, 43, 45, 49] |
| | Random 10 rules B ($\rho = 0.66$) | [6, 17, 20, 28, 29, 37, 40, 41, 47, 48] |
| Math | DPP 10 rules ($\rho = 0.40$) | [0, 4, 13, 26, 27, 31, 33, 38, 44, 45] |
| | Random 10 rules A ($\rho = 0.65$) | [0, 2, 11, 16, 17, 18, 27, 28, 34, 39] |
| | Random 10 rules B ($\rho = 0.61$) | [3, 4, 25, 20, 23, 13, 15, 24, 35, 39] |
| Code | DPP 10 rules ($\rho = 0.54$) | [2, 3, 13, 21, 28, 36, 37, 45, 46, 49] |
| | Random 10 rules A ($\rho = 0.59$) | [5, 7, 10, 13, 17, 19, 21, 26, 30, 34] |
| | Random 10 rules B ($\rho = 0.58$) | [2, 4, 8, 14, 16, 20, 23, 33, 37, 44] |

### A.7.9 GPT 10 UNCORRELATED RULES

Another straightforward rule generation method is to directly prompt GPT-4 to generate 10 uncorrelated rules and rely on its understanding of the correlation between the rules. We have explored this by using a similar rule generation prompt as in A.7.11, where we provide the task description and data description, but this time we request for 10 rules and added one sentence "make sure the rules are uncorrelated" to further require the independence of the rules. The 10 "uncorrelated" GPT rules are provided in Table 18 and 19 below. Following this, we rated the data according to the 10 GPT rules and calculated the rule correlation $\rho$ of the score vectors (in one *GPT-Uncorrelated* trial). We tested for Code and Math domain and got $\rho_{Code} = 0.65$ and $\rho_{Math} = 0.56$, both are significantly higher than DPP correlation values in Tabel 17. For Code, even random 10 rules selected from

a pool of 50 rules provide lower correlation than the 10 rules directly generated by GPT that are claimed to be "uncorrelated". This shows that our two-step approach—first generating enough rules to ensure diversity, followed by employing DPP on the rating vectors to select rules—is superior and also more task-specific.

| Index | Rule Description |
|---|---|
| 0 | Code Snippet Integrity: Select examples that contain complete and syntactically correct code snippets, avoiding those with partial or pseudo code which may confuse the model. |
| 1 | Language Diversity: Include examples in a variety of programming languages, ensuring that no single language dominates the dataset to promote versatility in code generation. |
| 2 | Comment Quality: Prioritize data that includes well-documented code with comments that clearly explain the logic and functionality. |
| 3 | Algorithmic Complexity: Choose examples that demonstrate a range of algorithmic solutions from basic to advanced. |
| 4 | Relevance to Modern Programming: Favor examples that utilize current libraries, frameworks, and features of programming languages. |
| 5 | Balanced Domain Representation: Ensure a balanced representation of code from different domains to prevent model bias. |
| 6 | Error Handling: Include examples that demonstrate robust error handling and debugging practices. |
| 7 | Executable Code: Select training examples where the code is functional and executable without errors. |
| 8 | Contextual Coherence: Ensure that the selected texts provide meaningful context that relates logically to the code. |
| 9 | Code Formatting and Style: Include examples that adhere to common coding standards and formats. |

Table 18: Generated 10 "uncorrelated" rules by GPT-4 for the Code domain.

| Index | Rule Description |
|---|---|
| 0 | Lexical Diversity Rule: Select data samples with a diverse vocabulary, especially those rich in mathematical terminology. Exclude texts with high repetition of common words and low occurrence of domain-specific terms. |
| 1 | Complexity and Structure Rule: Prioritize texts that exhibit complex sentence structures and logical argumentation, indicative of advanced reasoning skills. |
| 2 | Numerical Data Presence Rule: Include texts that contain numerical data, charts, or graphs, along with explanatory text that interprets or analyzes the numerical information. |
| 3 | Mathematical Concept Explanation Rule: Favor texts that explicitly explain mathematical concepts, theories, or problem-solving steps. |
| 4 | Contextual Relevance Rule: Select texts related to mathematical applications in real-world scenarios, such as physics problems or economics calculations. |
| 5 | Historical and Evolutionary Math Content Rule: Include content discussing the historical development and evolution of mathematical theories. |
| 6 | Cross-Disciplinary Integration Rule: Opt for texts that integrate mathematical concepts with other disciplines like science and engineering. |
| 7 | Error-free Mathematical Notation Rule: Ensure that the texts contain accurate and error-free mathematical notation wherever applicable. |
| 8 | Problem-Solving Narratives Rule: Select texts that include step-by-step problem-solving narratives or worked examples. |
| 9 | Cultural and Application Diversity Rule: Include texts that discuss the application of mathematics in various cultural and practical contexts. |

Table 19: Generated 10 "uncorrelated" rules by GPT-4 for the Code domain.

### A.7.10 USE GPT TO SELECT 10 UNCORRELATED RULES

In this part, we discuss a very similar setting to the previous section. However, instead of directly prompting GPT to generate 10 rules, we let GPT to replace the role of DPP and select 10 "uncorrelated" rules out of the 50-rule pool. First, in Table **??** below, we calculate the rule correlation similarly as in Table 17. We see again although we prompt GPT-4 to select "uncorrelated" rules, the rule-correlation of the selected 10 rules are still higher than our DPP-selected rules in Table 17. Moreover, we fine-tuned with the selected data and benchmarked the LLM performance. From the results in Table 21 and Table 22, we again see that it underperforms compared to our method.

| Domain | Method | Rule Indices |
|--------|--------|--------------|
| IMDB | GPT selected 10 rules ($\rho = 0.67$) | [0, 1, 4, 10, 13, 17, 25, 31, 40, 49] |
| Medical | GPT selected 10 rules ($\rho = 0.40$) | [0, 4, 7, 11, 15, 24, 29, 34, 42, 49] |
| Math | GPT selected 10 rules ($\rho = 0.56$) | [0, 3, 7, 11, 17, 24, 28, 38, 44, 48] |
| Code | GPT selected 10 rules ($\rho = 0.65$) | [0, 4, 9, 12, 16, 23, 29, 34, 43, 49] |

Table 20: Rule correlation and indices of the selected rules by DPP and random selection.

| Method | IMDB | Medical | | | |
|--------|------|---------|---|---|---|
| | SA accuracy | college medicine | professional medicine | medical genetics | Medical Average |
| GPT selected 10 rules | 51.6 | 21.9 | 42.2 | 24 | 29.4 |
| All 50 Rules | 51 | 23.1 | 42.6 | 23 | 29.6 |
| Random 10 Rules | 51.7 | 24 | 41.2 | 23.5 | 29.6 |
| DPP 10 Rules | **53.3** | **24.3** | **43** | **26** | **31.1** |

Table 21: IMDB & Medical fine-tuning on Pythia-1B, each using 20K selected data samples from SlimPajama.

| Method | Math | | | | Code | | | | |
|--------|------|---|---|---|------|---|---|---|---|
| | elementary | high school | college | Math Average | humaneval | mbpp | multiple-py | multiple-cpp | Code Average |
| GPT selected 10 rules | 42.5 | 40.3 | 36 | 39.6 | 40.8 | 43.4 | 42.8 | 50.3 | 44.3 |
| All 50 Rules | 41.8 | 40.7 | 33 | 38.5 | 43.9 | 43.4 | 46.6 | 49.1 | 45.8 |
| Random 10 Rules | 42.9 | 39.6 | 35 | 39.2 | 48.5 | 40.8 | 46.6 | 48.1 | 46 |
| DPP 10 Rules | **43.6** | 40.4 | **38** | **40.7** | **50.6** | **44.1** | 46.9 | **52.8** | **48.6** |

Table 22: Math & Code fine-tuning on Llama3-8B, each using 20K selected data samples from SlimPajama.

### A.7.11 PROMPTS AND GENERATED RULES

For brevity, we provide the templates for both the rule generation and rating prompts for the Math domain. To adapt these templates for other domains, replace terms specific to Math (such as "mathematical tasks" and "mathematical reasoning and analysis") with relevant terminology from the desired domain. We use GPT-4 to help us generate these task description and data descriptions.

**Rule Generation Prompts:**

Generate 50 specific rules for rating data from the training dataset (SlimPajama), in order to select a high-quality subset to train large language models that will improve their performance on mathematical tasks. The descriptions of the training data and the downstream task are provided below. The rules should focus on various aspects such as data quality, relevance, diversity, and other characteristics that would be beneficial for mathematical reasoning and analysis.

Description of training data:
<DATA_DESCRIPTION>

Description of downstream task:
<TASK_DESCRIPTION>

Requirements for the Rules:
Each rule should be concise and specific.
The rules could be basic text quality rules or task-related quality rules.
The rules should be written in clear, natural language and be easy to understand.

Now, please generate the 50 rules.

Figure 13: Example of a rule-generation prompt used to create 50 data rating rules for the Math domain.

**Rating Prompts:**

> We are training a language model using the SlimPajama dataset to improve performance on mathematical tasks. Evaluate the following example from SlimPajama dataset and assign a quality score between 0 and 1 (0 indicates the worst quality, and 1 indicates perfect quality) according to the provided rule:
> <RULE>
>
> Example:
> <DATA_SAMPLE>
>
> Respond only with a single float number.

Figure 14: Example of rule-rating prompt. Here we query the LLM to rate a single data sample based on a specific Math-related rule.

**Task and data descriptions:**

| Type | Description |
|---|---|
| SlimPajama data description | The SlimPajama dataset is a large-scale dataset. It is designed to be a compact, high-quality dataset curated for pre-training large language models. The dataset includes a diverse range of texts, sourced from various domains such as web pages, books, and academic articles, providing a rich and varied training corpus for developing robust and versatile language models. |
| IMDB task description | The IMDB review dataset, created by StanfordNLP, is a widely used dataset for sentiment analysis. It contains 50,000 highly polar movie reviews. Each review is labeled as either positive or negative, making it an ideal dataset for binary sentiment classification tasks. The dataset provides a challenging benchmark for evaluating the performance of sentiment analysis models. |
| Medical task description | The MMLU (Massive Multitask Language Understanding) includes three medical-related subsets: mmlu_college_medicine, mmlu_medical_genetics, and mmlu_professional_medicine. These subsets test a language model's understanding of general medical knowledge, genetic concepts, and advanced professional medical practices, respectively, through multiple-choice questions tailored to assess both foundational and specialized medical expertise. |
| Math task description | The MMLU (Massive Multitask Language Understanding) includes a range of subsets designed to evaluate language models across various academic subjects, including mathematics. The Math subsets specifically assess a model's capability to understand and solve mathematical problems. These are categorized into multiple difficulty levels—from elementary mathematics to college-level topics like abstract algebra. Each subset consists of multiple-choice questions that test different areas of mathematical knowledge, aiming to measure both basic arithmetic skills and more complex mathematical reasoning. This structure allows researchers to gauge a model's proficiency in mathematical logic and its application to solve real-world problems. |
| Code task description | The Code Generation LM Evaluation Harness, part of the BigCode project, is a framework designed to evaluate large language models (LLMs) on their ability to generate code. It provides a structured environment to assess the performance of these models across various programming tasks and languages. The harness supports automated evaluation metrics and facilitates benchmark comparisons, making it a valuable tool for researchers and developers aiming to enhance the code generation capabilities of LLMs. |

Table 23: Data descriptions of SlimPajama and task descriptions of four domains.

**Generated 50 rules for each of four domains:** Note that the IMDB rules here are used to select data for LLM training, whereas the IMDB rules in A.6.6 are used for data comparison in order to eventually calculate quality scores for the 50 IMDB reviews. Although similar, they are not the same set of rules.

| Index | Rule Description |
|---|---|
| 0 | Text Length: Be between 100 and 1000 words to match the typical length of IMDB reviews. |
| 1 | Sentiment Clarity: Clearly express either positive or negative sentiments. |
| 2 | Language Quality: Have fewer than 2 spelling or grammatical errors per 100 words. |

| Index | Rule Description |
|---|---|
| 3 | Language Focus: Be in English to maintain focus on the language of the target dataset. |
| 4 | Source Diversity: Be sourced evenly from web pages, books, and academic articles. |
| 5 | Tone Appropriateness: Minimize neutral tones as they are less useful for binary sentiment analysis. |
| 6 | Cultural Relevance: Discuss culturally significant topics relevant to a global English-speaking audience. |
| 7 | Language Style: Use informal, conversational language. |
| 8 | Sarcasm Avoidance: Avoid sarcasm to prevent misinterpretation by sentiment analysis models. |
| 9 | Subjectivity: Express opinions rather than just stating facts. |
| 10 | Emotional Expression: Express emotions to aid in sentiment understanding. |
| 11 | Redundancy Avoidance: Avoid redundancy and excessive similarity to other texts in the dataset. |
| 12 | Contemporary Relevance: Be from the past decade to ensure relevance. |
| 13 | Industry Relevance: Include mentions of movies, actors, or film industry terms. |
| 14 | Sentiment Indicators: Contain explicit sentiment indicators. |
| 15 | Sentence Complexity: Feature complex sentence structures. |
| 16 | Figurative Language: Use metaphors and similes. |
| 17 | Contextual Richness: Provide enough context to understand the sentiment on their own. |
| 18 | Jargon Avoidance: Avoid heavy use of irrelevant technical jargon. |
| 19 | Format Appropriateness: Avoid non-continuous formats like lists and tables. |
| 20 | Persuasiveness: Be persuasive, reflecting the tone often found in positive or negative reviews. |
| 21 | Genre Balance: Represent a balanced variety of genres (e.g., fiction, non-fiction, journalism). |
| 22 | Citation Minimization: Avoid being predominantly composed of citations or quotes. |
| 23 | Interactive Media Handling: Exclude interactive media texts unless they provide narrative value. |
| 24 | Structural Cohesion: Be cohesive and well-structured. |
| 25 | Offensive Content Avoidance: Avoid containing hate speech, excessive violence, or other offensive content. |
| 26 | Demographic Inclusivity: Discuss or be relevant to a variety of demographic groups. |
| 27 | Sentiment Extremity: Express strong sentiments, either positive or negative. |
| 28 | Colloquial Language: Mimic spoken language, as often found in movie reviews. |
| 29 | Descriptive Nature: Avoid being purely descriptive and lack subjective opinions. |
| 30 | Historical Context: Include historical references only if they enhance the sentiment or narrative. |
| 31 | Plagiarism Avoidance: Be free from plagiarism. |
| 32 | Domain-Specific Language: Contain relevant film and media terms. |
| 33 | User-Generated Content: Include user-generated content such as blogs and user reviews. |
| 34 | Narrative Emphasis: Be narrative-driven, resembling the storytelling found in reviews. |
| 35 | Error Avoidance: Avoid formatting or data errors. |
| 36 | Topical Relevance: Discuss topics commonly found in movie reviews such as plot, acting, and direction. |
| 37 | Satire Handling: Avoid satire unless it is clearly marked or well-known. |
| 38 | Subject Line Clarity: Have moderate and descriptive subject lines. |
| 39 | Outdated Content Avoidance: Avoid containing outdated societal views or terminologies. |
| 40 | Regional Representation: Represent various English dialects and regional variations. |
| 41 | Emotional Variability: Exhibit a range of emotions from joy to sadness, to anger. |
| 42 | Controversial Topic Inclusion: Include discussions on controversial topics if they enhance sentiment understanding. |
| 43 | Generalization Avoidance: Avoid making broad generalizations without substantiation. |
| 44 | Source Reliability: Be from reliable and reputable sources. |
| 45 | Uniqueness: Be unique with no duplicates in the dataset. |
| 46 | Formality Variance: Include a variety of formality levels, particularly matching the informal style of many movie reviews. |
| 47 | Impactful Sentences: Contain emotionally resonant sentences critical for sentiment analysis. |
| 48 | Engagement: Be engaging and likely to provoke reader reactions. |
| 49 | Visual Storytelling: Include vivid descriptions akin to visual storytelling in movies. |

Table 24: Generated 50 rules for the IMDB domain.

| Index | Rule Description |
|---|---|
| 0 | Relevance to Medical Topics: Include texts that contain medical terminology or discuss medical topics. |
| 1 | Exclusion of Non-Medical Content: Exclude texts that do not pertain to health, medicine, or biological sciences. |
| 2 | Clarity of Medical Information: Select texts where medical information is clearly explained and easy to understand. |
| 3 | Accuracy of Medical Content: Ensure texts contain medically accurate information, verified against reputable medical sources. |
| 4 | Diversity of Medical Subfields: Include texts covering a range of medical fields such as genetics, anatomy, pharmacology, and pathology. |
| 5 | Contemporary Relevance: Prefer texts discussing current medical practices and technologies over outdated treatments. |

| Index | Rule Description |
|---|---|
| 6 | Technical Depth: Include texts with a deep, technical discussion of medical topics suitable for professional medicine. |
| 7 | Exclusion of Ambiguous Content: Avoid texts with ambiguous or unclear medical claims or data. |
| 8 | Citation of Sources: Select texts that cite reputable medical journals or textbooks. |
| 9 | Grammar and Spelling: Ensure texts are free from grammatical errors and spelling mistakes. |
| 10 | Use of Professional Language: Prefer texts that utilize professional medical jargon correctly. |
| 11 | Inclusion of Case Studies: Include texts that discuss medical case studies or clinical trials. |
| 12 | Representation of Rare Diseases: Ensure inclusion of texts discussing rare or less common diseases. |
| 13 | Coverage of Ethical Considerations: Include texts discussing ethical considerations in medical practice and research. |
| 14 | Language Diversity: Include texts in multiple languages relevant to global medical practice. |
| 15 | Patient Education Focus: Include texts aimed at patient education that explain medical conditions and treatments clearly. |
| 16 | Statistical Data Presentation: Prefer texts that present medical data and statistics clearly. |
| 17 | Illustration of Medical Procedures: Include texts with detailed descriptions or illustrations of medical procedures. |
| 18 | Pharmacological Content: Include texts discussing drug mechanisms, interactions, side effects, and benefits. |
| 19 | Genetic Concepts Coverage: Ensure texts covering genetic concepts are detailed and accurate. |
| 20 | Medical Research Updates: Include texts with the latest research findings in the medical field. |
| 21 | Interdisciplinary Approach: Select texts that integrate medical knowledge with other sciences like biochemistry or physics. |
| 22 | Historical Medical Milestones: Include texts discussing historical advancements in medicine. |
| 23 | Medical Guidelines and Protocols: Include texts that detail medical guidelines, protocols, or standard operating procedures. |
| 24 | Interviews with Medical Professionals: Include interviews or discussions with recognized experts in the medical field. |
| 25 | Patient Case Confidentiality: Exclude texts that potentially breach patient confidentiality or privacy. |
| 26 | Texts from Medical Conferences: Include content from recent medical conferences or symposiums. |
| 27 | Exclusion of Pseudoscience: Strictly exclude texts promoting unverified or pseudoscientific claims. |
| 28 | Clinical Pathway Discussions: Include texts discussing clinical decision-making processes and pathways. |
| 29 | Medical Device Descriptions: Include texts that describe the use and innovation of medical devices. |
| 30 | Nutritional and Lifestyle Medicine: Include texts discussing the impact of nutrition and lifestyle on health. |
| 31 | Pediatric Medicine Coverage: Ensure texts covering pediatric medicine are included. |
| 32 | Mental Health Discussions: Include texts that address various aspects of mental health care. |
| 33 | Healthcare Policy Analysis: Include texts analyzing healthcare policies and their implications. |
| 34 | Disease Prevention Focus: Include texts focused on disease prevention strategies and methods. |
| 35 | Surgical Techniques Description: Prefer texts that detail surgical procedures and techniques. |
| 36 | Medical Training and Education: Include texts related to medical training and education methods. |
| 37 | Veterinary Medicine: Include texts on veterinary medicine where relevant to comparative medicine. |
| 38 | Environmental Health Issues: Include texts discussing the impact of environmental factors on health. |
| 39 | Bioinformatics Data Handling: Include texts discussing the handling and analysis of bioinformatics data. |
| 40 | Medical Imaging Techniques: Include texts discussing modern medical imaging techniques and their applications. |
| 41 | Cultural Competence in Healthcare: Include texts that discuss cultural considerations in healthcare provision. |
| 42 | Global Health Challenges: Include texts discussing global health issues and strategies. |
| 43 | Emergency Medicine Protocols: Include texts detailing protocols and procedures in emergency medicine. |
| 44 | Health Insurance Systems: Include texts discussing different health insurance systems and policies. |
| 45 | Medical Ethics Case Studies: Include case studies discussing medical ethics dilemmas and resolutions. |
| 46 | Integrative Medicine Approaches: Include texts on integrative approaches combining traditional and modern medicine. |
| 47 | AI and Machine Learning in Medicine: Include discussions on the application of AI and machine learning in medical contexts. |
| 48 | Telemedicine and Remote Care: Include texts on the advancements and challenges in telemedicine. |
| 49 | Healthcare Accessibility and Equity: Include texts discussing issues of accessibility and equity in healthcare. |

Table 25: Generated 50 rules for the Medical domain.

| Index | Rule Description |
|---|---|
| 0 | Mathematical Keywords: Prioritize texts containing keywords related to mathematics such as 'algebra', 'calculus', 'geometry', 'equations', 'theorems', etc. |
| 1 | Problem Statements: Include examples that present mathematical problems or puzzles. |
| 2 | Solution Explanations: Select texts that not only present problems but also explain solutions step-by-step. |
| 3 | High-Quality Sources: Favor texts sourced from academic articles, educational websites, and textbooks over general web pages. |
| 4 | Symbolic Representation: Ensure the presence of mathematical symbols and expressions formatted in LaTeX or similar markup languages. |
| 5 | Advanced Topics Coverage: Include texts that cover advanced mathematical topics such as differential equations, statistics, and abstract algebra. |
| 6 | Logical Structuring: Texts should demonstrate clear logical structuring, particularly in argumentation and problem-solving. |

| Index | Rule Description |
|---|---|
| 7 | Historical Context: Include content that provides historical context or development of mathematical theories and applications. |
| 8 | Data Sets and Examples: Prioritize texts that include real-world data sets or examples where mathematical principles are applied. |
| 9 | No Misconceptions: Exclude texts containing mathematical misconceptions or common errors unless they are being corrected. |
| 10 | Illustrations and Diagrams: Include texts with clear diagrams, graphs, and illustrations that aid mathematical understanding. |
| 11 | Proofs and Theorems: Include detailed explanations of proofs and discussions of theorems. |
| 12 | Mathematics in Technology: Include examples that link mathematics with its applications in technology and engineering. |
| 13 | Interdisciplinary Links: Select texts that illustrate the application of mathematics in other scientific disciplines like physics and chemistry. |
| 14 | Question and Answer Format: Include texts that follow a question and answer format, especially for complex mathematical concepts. |
| 15 | Exclusion of Irrelevant Content: Exclude texts that are primarily non-mathematical in nature, such as pure narrative or opinion pieces. |
| 16 | Mathematical Definitions: Include texts that provide clear definitions of mathematical terms and concepts. |
| 17 | Tutorial Style: Select tutorial-style texts that are aimed at teaching or explaining mathematical concepts. |
| 18 | Accuracy of Content: Exclude any text with factual inaccuracies related to mathematics. |
| 19 | Age-Appropriate Content: Select content that is appropriate for the educational level, from elementary to college-level mathematics. |
| 20 | Challenge Level: Include texts with varying levels of difficulty to ensure a range of challenges in problem-solving. |
| 21 | Language Clarity: Ensure the text uses clear and precise language appropriate for teaching or explaining mathematics. |
| 22 | Cultural Diversity: Include mathematical content from diverse cultural backgrounds to promote inclusivity. |
| 23 | Recency of Content: Prioritize recent texts that reflect the current state of mathematical education and theory. |
| 24 | Real-World Applications: Select texts that discuss the application of mathematical concepts in real-world scenarios. |
| 25 | Peer-Reviewed Sources: Favor texts extracted from peer-reviewed academic journals and conferences. |
| 26 | Multiple Perspectives: Include texts that present multiple perspectives or methods for solving a single mathematical problem. |
| 27 | Step-by-Step Guides: Prioritize texts that provide step-by-step guides to solving mathematical problems. |
| 28 | Integration of Tools: Include texts that discuss or utilize mathematical tools and software. |
| 29 | Variety of Formats: Include a variety of text formats such as articles, essays, and problem sets. |
| 30 | Consistency in Terminology: Ensure consistency in mathematical terminology across the selected texts. |
| 31 | Explanatory Footnotes: Include texts that make use of footnotes or side-notes to explain complex terms or provide additional context. |
| 32 | Interactive Elements: Select texts that include or suggest interactive elements like quizzes or interactive diagrams. |
| 33 | Avoid Redundancy: Avoid texts that are redundant in content, especially if they do not add new information or perspective. |
| 34 | Mathematical Puzzles: Include texts that feature mathematical puzzles and games to enhance problem-solving skills. |
| 35 | Comparative Analyses: Select texts that involve comparative analyses of different mathematical methods or theories. |
| 36 | Language Models and Mathematics: Include texts discussing the intersection of language processing models and mathematics. |
| 37 | Excerpts from Lectures: Include transcribed excerpts from academic lectures on mathematics. |
| 38 | Mathematical Narratives: Include narratives that weave mathematical concepts into broader storylines or real-life applications. |
| 39 | Authoritative Authors: Prioritize texts authored by well-regarded mathematicians or educators. |
| 40 | Exclusion of Vague Language: Avoid texts that use vague or ambiguous language when explaining mathematical concepts. |
| 41 | Feedback Loops: Include texts that describe the importance of feedback loops in mathematical learning. |
| 42 | Error Analysis: Include texts that focus on error analysis in mathematical calculations or theories. |
| 43 | Cross-Referencing: Favor texts that cross-reference other works or theories effectively. |
| 44 | Mathematical Software Tutorials: Include tutorials or guides on using mathematical software. |
| 45 | Engagement Metrics: Favor texts that have historically engaged readers or viewers, indicating quality and interest. |
| 46 | Student Contributions: Include texts written by students, which can provide fresh perspectives and innovative approaches. |
| 47 | Reviews and Critiques: Select texts that review or critique mathematical theories or textbooks. |
| 48 | Accessibility Features: Include texts that are accessible to people with disabilities, such as those formatted for screen readers. |
| 49 | Alignment with Curriculum: Ensure that the content aligns well with standard mathematical curriculums at various educational levels. |

Table 26: Generated 50 rules for the Math domain.

| Index | Rule Description |
|---|---|
| 0 | Syntax Highlighting: Include texts that contain syntax highlighting or structured code comments. |
| 1 | Grammar Quality: Exclude texts with excessive spelling and grammatical errors. |

| Index | Rule Description |
|---|---|
| 2 | Programming Keywords: Prioritize samples containing programming language keywords and constructs. |
| 3 | Language Focus: Exclude texts that are predominantly non-English unless they are code snippets. |
| 4 | Concept Explanation: Select texts with clear, concise explanations of programming concepts. |
| 5 | Reputable Sources: Prioritize texts from reputable sources like well-known programming blogs and documentation sites. |
| 6 | Minimum Length: Exclude texts that contain less than 50 words as they may not provide sufficient context. |
| 7 | Best Practices: Include examples that demonstrate best coding practices. |
| 8 | Language Diversity: Prioritize texts that include diverse programming languages covered in the BigCode project. |
| 9 | Error Solutions: Select texts that provide examples of common programming errors and their solutions. |
| 10 | Technique Comparison: Include texts with comparative discussions of different coding techniques or tools. |
| 11 | Current Practices: Exclude texts with outdated or deprecated coding practices. |
| 12 | Algorithm Explanation: Prioritize texts that include algorithm explanations with code snippets. |
| 13 | Duplication Check: Exclude samples that are heavily duplicated within the dataset. |
| 14 | API Usage: Select samples that demonstrate use of APIs from well-known software libraries. |
| 15 | Multi-Language Code: Include texts with embedded code in multiple programming languages. |
| 16 | Development Paradigms: Prioritize texts that discuss software development paradigms (e.g., object-oriented programming). |
| 17 | Relevance Check: Exclude non-relevant texts like purely historical accounts of programming without technical details. |
| 18 | Decision Context: Include texts that provide context on why certain coding decisions are made. |
| 19 | Annotated Code: Prioritize texts that contain code with annotations explaining each part of the code. |
| 20 | Step-by-Step Code: Include samples where code is broken down into step-by-step explanations. |
| 21 | Non-Promotional: Exclude texts that are purely promotional or sales-focused. |
| 22 | Debugging Techniques: Select texts that discuss debugging techniques with code examples. |
| 23 | Tool Comparison: Include texts that compare different programming tools or environments. |
| 24 | Technical Focus: Prioritize articles or excerpts from technical books that focus on programming. |
| 25 | Academic Pseudo-Code: Include texts from academic papers that contain pseudo-code or algorithms. |
| 26 | Content Density: Exclude texts that are excessively verbose without substantive content. |
| 27 | Optimization Tips: Prioritize texts that provide insights into code optimization. |
| 28 | Advanced Topics: Include texts that cover advanced programming topics like concurrency or security. |
| 29 | Architecture Patterns: Select texts that discuss architectural patterns with code examples. |
| 30 | Programming Paradigms: Prioritize examples that demonstrate functional or logic-based programming. |
| 31 | Technical Emphasis: Exclude samples that focus solely on non-technical aspects of IT projects. |
| 32 | Quality Solutions: Include forum and Q&A entries with high-quality code solutions. |
| 33 | Documentation Inclusion: Select project documentation and readme files that include example usage of code. |
| 34 | Commented Code: Prioritize texts with code that includes comprehensive inline comments. |
| 35 | Jargon Balance: Exclude texts with a high density of technical jargon unless accompanied by clear explanations or code. |
| 36 | Executable Snippets: Include code snippets that are functional and can be executed without modifications. |
| 37 | Complexity Discussion: Prioritize texts that explain the computational complexity of algorithms with examples. |
| 38 | Integration Showcase: Include texts that showcase the integration of different technologies or languages. |
| 39 | Version Control: Select samples that explain version control practices with code snippets. |
| 40 | Cross-Platform Coding: Prioritize texts that discuss cross-platform coding challenges and solutions. |
| 41 | Interactive Tutorials: Include interactive coding tutorials or walkthroughs. |
| 42 | Proprietary Code: Exclude any samples containing proprietary code without proper authorization. |
| 43 | Scalability Focus: Select examples that discuss the scalability of code or systems. |
| 44 | Accessibility Coding: Prioritize samples that address coding for accessibility or internationalization. |
| 45 | Performance Analysis: Include texts that analyze the performance of different coding approaches. |
| 46 | Content Relevance: Exclude texts that mix code with irrelevant images or multimedia that don't add educational value. |
| 47 | Ethical Coding: Prioritize texts that discuss ethical considerations in programming. |
| 48 | Tool Usage: Include examples of how to use popular development tools and environments through coding tutorials. |
| 49 | Project Scope: Select texts that clearly define the scope and objectives of programming projects. |

Table 27: Generated 50 rules for the Code domain.

