# OpenReview forum: "Rule-Based Rating and Selection of LLM Training Data"
_ICLR.cc/2025/Conference — Submitted to ICLR 2025_

### Official Review · Reviewer_9u4d · 2024-10-29

**Soundness:** 3
**Presentation:** 3
**Contribution:** 3
**Rating:** 6
**Confidence:** 3

**Summary:**

The paper solves the problem of choosing a subset of training data with better quality that can help in better finetuning or pre-training of the LLMs. The paper removes the need for human intervention in rule generation by automatically generating the rules using the LLM model itself. It also proposes a method to filter out rules from a large rule set to obtain diverse and independent set of rules. Overall it provides an automated solution to obtain a subset of training data for better training without any human in the loop.

**Strengths:**

1. The paper reduces the cost by removing the human experts for the creation of rules.

2. Paper has good amount of experiments that support the claims made in the paper.

3. The paper also introduces a robust rule-evaluation metric that promotes diverse and independent set of rules.

**Weaknesses:**

1. The paper is quite complex and might take some amount of effort to correctly implement.

2. It depends heavily on LLMs such as GPT-4 which might not be an efficient approach for the resource constraint scenarios and require huge computational costs.

3. The cost of human in the loop is replaced by the cost of performing inference on the LLMs such as GPT-4 that should be considered, this is one major drawback of the paper and there should be an ablation study over that. There is a trade-off between cost of human in the loop and the cost of executing an LLM that needs to be explored.

**Questions:**

**See weaknesses.**

---

> ### Author Response · Authors · 2024-11-18
> **Response to Reviewer 9u4d (Part 1/2)**
>
> We appreciate the valuable feedback and below are our responses (all the line numbers and sections are referred to the new pdf version):
>
> **1. (a) Paper too complex, and (b) Implementation:**
>
> (a) Because our pipeline is entirely automated and end-to-end, and it is a complete pipeline starting from automatic rule generation all the way to the final data selection for downstream tasks, it might look complex to some readers. Nonetheless, the idea is simple, within an LLM-as-a-judge setting (which means using powerful LLMs to rate data quality for selection, and is one of the most popular methods now for quality data selection [1][2][3][4][5][6][7][8][9]), we **investigate what quality aspects we should consider for data rating**. Our strategy is to **first generate a large rule pool to ensure we cover enough aspects, and then remove the correlated ones to make the rules more even/balanced**. We proposed the quantitative measure to assess this “correlation”, and used DPP as the algorithm to achieve this selection of uncorrelated rules. We hope this clarifies the essential idea of our method.
>
> (b) Indeed our implementation needs the rating and selection process, but these are inevitable under the LLM-as-a-judge setting [1][2][3][4][5][6][7][8][9]. Our method only adds the rule generation and selection step to standard LLM-as-a-judge implementation.
>
> **2. Heavily depends on GPT-4:**
>
> For LLM-as-a-judge, powerful LLM such as GPT-4 is often used as the judge [1][2][3][4][5][6][7][8][9]. Moreover, due to efficiency, we use Llama3-8B-Instruct as the rater, which is a lighter model than GPT-4. **We only used GPT-4 for rule generation, which is done by only 1 prompt**. Our paper focuses on solving the issue within the LLM-as-a-judge setting, which is the challenge of designing and aggregating the rating aspects (rules). When the ratings are available, our DPP selection is extremely fast. For more details on the run time, please refer to the paper (Line 209-215 and 807-818).
>
> If the reviewer worries about the reliance on GPT for rule generation: we have **provided the comparison against using Claude as the rule generator in Section A.6.5(Line 988-1018). We use PCA to visualize the embedding (generated using sentence-transformer) of the two sets of rules**. It is observed that the two clusters of rules generated by GPT and Claude completely overlap, demonstrating no distinct separation. Additionally, **we have further computed the Wasserstein distance as a quantitative evaluation of the distribution differences between GPT rules and Claude rules**. Specifically, we compute the distance between the GPT rules and Claude rules. Then we randomly split GPT rules into two parts and computed their Wasserstein distance and similarly for the Claude rules. By comparing these values we found no clear distribution bias when switching from one rule generator to the other. Please see Table 4 (Line 1022-1032 of most recent version).
>
> Moreover, **if the reviewer concerns about the general bias brought by LLMs (including GPT and Claude), to address the concern, in Line 1034-1042, compared to human-generated rules, we prompt GPT-4 to generate ethical and safety rules and compare them with the human-designed rules in Collective Constitutional AI[1]**. We asked 3 authors who have not seen the constitutions to distinguish the rules blindly. We get an average accuracy of 20.7%, suggesting it is indeed hard to distinguish between rules generated by GPT and those designed by humans. All these discussions underscore the potential of GPT as a reliable rule generator that is capable of producing rules that are comparable to those crafted by human experts. We thank the reviewer for pointing this out and we hope these in some aspect address the reviewer’s concern about GPT-rule-generation bias.

---

> ### Author Response · Authors · 2024-11-18
> **Response to Reviewer 9u4d (Part 2/2)**
>
> **3. Cost of human in the loop:**
>
> **As mentioned above, the entire LLM-as-a-judge is proposed to replace human raters and has been commonly adopted by researchers**. In our case, we further address the bias of the rules designed and aggregated by humans. Our fully automated method eliminates the need for **repeated human intervention with each new task, which is not only costly but also difficult to integrate into certain workflows**, such as composite reward models in RLHF. Human in the loop is usually more expensive than using automatic models [7][8][9][10]. Moreover, when humans attempt to balance rules to minimize bias, they depend on their semantic understanding, yet the same rules can exhibit different rating correlations across datasets. For example, in QuRating, the designed rules unexpectedly showed high correlations, potentially introducing bias and redundancy in data quality ratings, and obstructing rule customization or re-weighting. In contrast, our method uses to the score representations of rules, which is guaranteed to be data-dependent. As detailed in Section A.7.9 (Lines 1481) and A.7.10 (Table 19, Line1537-1541), we analyzed the correlation of the “uncorrelated” rules generated or selected by GPT and found that their actual rating scores were significantly more correlated than those selected via the DPP method.
>
> In summary, we are grateful to the reviewers for highlighting these points, and we hope our detailed discussion and additional experiments potentially addressed the concerns. We would appreciate it if the reviewers could consider increasing the score based on these updates!
>
> [1] Weizhe Yuan, Richard Yuanzhe Pang, Kyunghyun Cho, Sainbayar Sukhbaatar, Jing Xu, and Jason Weston. Self-rewarding language models. arXiv preprint arXiv:2401.10020, 2024
>
> [2] Yuntao Bai, Saurav Kadavath, Sandipan Kundu, Amanda Askell, Jackson Kernion, Andy Jones, Anna Chen, Anna Goldie, Azalia Mirhoseini, Cameron McKinnon, et al. Constitutional ai: harm-lessness from ai feedback. 2022. arXiv preprint arXiv:2212.08073, 2022.
>
> [3] Alexander Wettig, Aatmik Gupta, Saumya Malik, and Danqi Chen. Qurating: Selecting high-quality
> data for training language models. arXiv preprint arXiv:2402.09739, 2024.
>
> [4]Zhiqing Sun, Yikang Shen, Qinhong Zhou, Hongxin Zhang, Zhenfang Chen, David Cox, Yiming Yang, and Chuang Gan. Principle-driven self-alignment of language models from scratch with minimal human supervision. Advances in Neural Information Processing Systems, 36, 2024.
>
> [5] Tong Mu, Alec Helyar, Johannes Heidecke, Joshua Achiam, Andrea Vallone, Ian Kivlichan, Molly Lin, Alex Beutel, John Schulman, and Lilian Weng. Rule based rewards for language model safety.
>
> [6] Pat Verga, Sebastian Hofstatter, Sophia Althammer, Yixuan Su, Aleksandra Piktus, Arkady Arkhangorod-sky, Minjie Xu, Naomi White, and Patrick Lewis. Replacing judges with juries: Evaluating llm generations with a panel of diverse models. arXiv preprint arXiv:2404.18796, 2024.
>
> [7] Harrison Lee, Samrat Phatale, Hassan Mansoor, Thomas Mesnard, Johan Ferret, Kellie Ren Lu, Colton Bishop, Ethan Hall, Victor Carbune, Abhinav Rastogi, et al. Rlaif vs. rlhf: Scaling reinforcement learning from human feedback with ai feedback. In Forty-first International Conference on Machine Learning
>
> [8] Zheng, Lianmin, et al. "Judging llm-as-a-judge with mt-bench and chatbot arena." Advances in Neural Information Processing Systems 36 (2023): 46595-46623.
>
> [9] Huang, Hui, et al. "An empirical study of llm-as-a-judge for llm evaluation: Fine-tuned judge models are task-specific classifiers." arXiv preprint arXiv:2403.02839 (2024).
>
> [10] Ziegler, Daniel M., et al. "Fine-tuning language models from human preferences." arXiv preprint arXiv:1909.08593 (2019).

---

> ### Author Response · Authors · 2024-11-24
> **Gentle Reminder**
>
> We sincerely appreciate the insightful comments provided by Reviewer 9u4d! To address the reviewer's concern, we have conducted additional experiments and added the corresponding analysis and discussions. This is a friendly reminder that the deadline of the rebuttal period is approaching. We would really appreciate it if the reviewer could take a look at our responses at their earliest convenience. We hope that the additional experiments and discussions address the reviewer’s concerns effectively and would be deeply grateful if the reviewer could consider raising the score!

---

> > ### Comment · Reviewer_9u4d · 2024-11-25
> > **Rebuttal response**
> >
> > Thanks for the response. Increased score from 5 --> 6.

---

> > > ### Author Response · Authors · 2024-11-26
> > >
> > > Thank you for raising the score!

---

### Official Review · Reviewer_ZVjx · 2024-11-01

**Soundness:** 4
**Presentation:** 4
**Contribution:** 4
**Rating:** 6
**Confidence:** 4

**Summary:**

The paper introduces an automated rule-based framework for selecting high-quality training data for large language models (LLMs) without human intervention. The framework utilizes LLMs to generate a diverse set of rules for data evaluation, addressing limitations in existing methods that rely on human-designed heuristics. It employs a determinantal point process (DPP) to select a subset of rules that are orthogonal, effectively reducing redundancy and bias in data selection. This method aims to improve data quality by rating and selecting data samples with the highest average scores, making it versatile across pre-training, fine-tuning, and RLHF tasks. The paper validates this approach through two evaluations: comparison with ground-truth ratings and performance assessment of LLMs trained with selected data, across domains like IMDB, Medical, Math, and Code. The results show that the DPP rule-based rating outperforms other methods, confirming its effectiveness in enhancing both rating accuracy and model performance.

**Strengths:**

1. The paper presents an automated rule-based framework for selecting high-quality LLM training data without human intervention. The model generates diverse rules for data evaluation, effectively eliminating human bias.
2. The method is flexible across various scenarios, including pre-training, SFT, and RLHF, and can be adapted to specific domains by modifying rule-generation prompts.
3. Extensive experiments covering different downstream tasks validate the approach. The detailed appendices further enhance the reliability of the results.
4. The rule evaluation metric focuses on score similarity rather than direct scoring, enabling measurement of rule diversity and adaptability to multiple domains and tasks.

**Weaknesses:**

1. Although the paper verifies the method’s effectiveness through extensive experiments, the proposed approach lacks significant innovation. Diversity-based selection and DPP for data selection are relatively common in prior work [1]. This paper’s difference lies in selecting rules rather than data directly. Citing related work and clearly distinguishing this approach from previous studies would strengthen the paper’s contribution.

[1] Yang Y, Wang H, Wen M, et al. P3: A Policy-Driven, Pace-Adaptive, and Diversity-Promoted Framework for Optimizing LLM Training[J]. arXiv preprint arXiv:2408.05541, 2024.

2. The approach requires manual adjustment of prompts for different tasks and datasets, meaning that the effectiveness of the rules is limited by the accuracy of the prompts. It is also unclear how the descriptions of data and downstream tasks are generated. Additionally, the value of hyperparameter r is constrained by the value of R, and the paper does not clarify how to choose the optimal r.

3. Rule quality is evaluated only by orthogonality/correlation, with no evidence that unrelated rule combinations are the most effective. Additional metrics would offer a more comprehensive assessment.

I would be glad to improve my score if my concerns could be addressed.

**Questions:**

None

---

> ### Author Response · Authors · 2024-11-18
> **Response to Reviewer ZVjx (Part 1/2)**
>
> We appreciate the valuable feedback and below are our responses (all the line numbers and sections are referred to the new pdf version):
>
> **1. Citation to P3:**
>
> We appreciate the reviewer for bringing this paper to our attention. According to the dates, this paper is considered a concurrent work. Moreover, the new version of the Arxiv is three weeks later than our ICLR submission. However, as suggested by the reviewer, we have **added the citation and the discussion in Line 818-825 to compare this with our method**. The only thing our work shares with theirs is using DPP as the diversity tool. Particularly, the authors in P3 also use DPP to perform data selection, but directly applied to the data itself. Nonetheless, the approach to directly perform data selection using DPP requires the computation based on the kernel matrix with a dimension equal to the number of samples, which is usually huge (could be millions) in the context of LLM data, where for our application of DPP on the rules, the dimension is the number of rules (typically dozens). Moreover, while DPP data selection inherently ensures data diversity, this is just one aspect of data quality. In contrast, our rule-based approach guarantees diversity and balance between the rules, i.e. the rating quality aspects. Our framework assesses multiple aspects of data quality, ensuring a more comprehensive and robust selection process. We think is it helpful to add the citation and add this discussion to compare the crucial differences, and thank the reviewer for this valuable suggestion!
>
> **2. (a) Rating prompts with task descriptions, and (b) Optimal r**
>
> (a) For the setting of LLM-as-a-judge, a rating prompt is inevitable [1][2][3][4][5][6][7][8][9]. **There are no universally standard prompts and our rating prompt is already a refined version** of the prompts used in previous works such as Self-Rewarding[1] and Constitutional AI[2], by adding data and task descriptions. We **added the clarification in Line 1565 of the paper that the data and task descriptions are generated with the assistance of GPT-4**. It is quite common to use human-designed prompts or GPT-generated prompts for LLMs (for example see [1][2][3][4][5]). We appreciate the reviewer for pointing this out!
>
> (b) For the optimal choice of r, we first emphasize that studying the optimal r essentially is the question of how many aspects we should consider for data quality selection, which is a very challenging question. Our **experiments in Section 5 (Evaluation B) explore the optimal r (please see Figure 3 and further Figure 7)**. We observed that optimal r is not near the boundaries but rather **around 10-15. This motivates our choice of r=10** in the paper. For fine-tuning experiments, it is impossible for us to have enough GPT resources to investigate exactly the optimal r. However, our experiments of using all 50 rules, and variation of r=20 in A.7.5 (Line 1318-1330), all further confirm our hypothesis that the optimal r is not near the boundaries. We have emphasized the motivation for the choice of r=10 in Line 361.

---

> ### Author Response · Authors · 2024-11-18
> **Response to Reviewer ZVjx (Part 2/2)**
>
> **3. Other rule evaluations:**
>
> Thanks for mentioning this insightful question. **In fact, evaluating or determining the optimal choice of data quality aspects (rules) is extremely challenging**[3][4][5]. As far as we know, all the previous works rely on human-designed rules but struggle with how to design, select, and weigh the rules. We are the first to propose a quantitative criterion for data quality aspects selection. While this criterion may not be comprehensive, it is essential. As discussed in the paper, rule correlations could disrupt balance and evenness, resulting in biased ratings toward certain aspects. In fact, in some domains, such as the **IMDB, we have observed that the Pearson correlation of the rule orthogonality with the rule accuracy to be as high as more than 0.6 (please see Figure 6)**, indicating that the benefits of orthogonality may be greater than anticipated. We concur with the reviewer that this is just a starting point, and a more thorough evaluation of rules is necessary, pointing to future areas of research.
>
> In summary, we thank the reviewer for mentioning the related paper and asking insightful questions! We hope our detailed discussion could help address the concerns and would appreciate if the reviewer could improve the score!
>
> [1] Weizhe Yuan, Richard Yuanzhe Pang, Kyunghyun Cho, Sainbayar Sukhbaatar, Jing Xu, and Jason Weston. Self-rewarding language models. arXiv preprint arXiv:2401.10020, 2024
>
> [2] Yuntao Bai, Saurav Kadavath, Sandipan Kundu, Amanda Askell, Jackson Kernion, Andy Jones, Anna Chen, Anna Goldie, Azalia Mirhoseini, Cameron McKinnon, et al. Constitutional ai: harm-lessness from ai feedback. 2022. arXiv preprint arXiv:2212.08073, 2022.
>
> [3] Alexander Wettig, Aatmik Gupta, Saumya Malik, and Danqi Chen. Qurating: Selecting high-quality
> data for training language models. arXiv preprint arXiv:2402.09739, 2024.
>
> [4]Zhiqing Sun, Yikang Shen, Qinhong Zhou, Hongxin Zhang, Zhenfang Chen, David Cox, Yiming Yang, and Chuang Gan. Principle-driven self-alignment of language models from scratch with minimal human supervision. Advances in Neural Information Processing Systems, 36, 2024.
>
> [5] Tong Mu, Alec Helyar, Johannes Heidecke, Joshua Achiam, Andrea Vallone, Ian Kivlichan, Molly Lin, Alex Beutel, John Schulman, and Lilian Weng. Rule based rewards for language model safety.
>
> [6] Pat Verga, Sebastian Hofstatter, Sophia Althammer, Yixuan Su, Aleksandra Piktus, Arkady Arkhangorod-sky, Minjie Xu, Naomi White, and Patrick Lewis. Replacing judges with juries: Evaluating llm generations with a panel of diverse models. arXiv preprint arXiv:2404.18796, 2024.
>
> [7] Harrison Lee, Samrat Phatale, Hassan Mansoor, Thomas Mesnard, Johan Ferret, Kellie Ren Lu, Colton Bishop, Ethan Hall, Victor Carbune, Abhinav Rastogi, et al. Rlaif vs. rlhf: Scaling reinforcement learning from human feedback with ai feedback. In Forty-first International Conference on Machine Learning
>
> [8] Zheng, Lianmin, et al. "Judging llm-as-a-judge with mt-bench and chatbot arena." Advances in Neural Information Processing Systems 36 (2023): 46595-46623.
>
> [9] Huang, Hui, et al. "An empirical study of llm-as-a-judge for llm evaluation: Fine-tuned judge models are task-specific classifiers." arXiv preprint arXiv:2403.02839 (2024).

---

> > ### Comment · Reviewer_ZVjx · 2024-11-24
> >
> > Thank you for your detailed response and additional explanations included in the current version. These updates have resolved my concerns and I have increased my score accordingly.

---

> > > ### Author Response · Authors · 2024-11-24
> > >
> > > We really appreciate the insightful comments and questions! Thank you for raising the score!

---

### Official Review · Reviewer_1T3n · 2024-11-03

**Soundness:** 3
**Presentation:** 3
**Contribution:** 3
**Rating:** 6
**Confidence:** 4

**Summary:**

This paper introduces an automated method to improve the quality of training data for large language models (LLMs). Instead of relying on human-made rules, the authors use GPT-4 to automatically generate a wide range of data quality rules based on the specific task and dataset. They then apply Determinantal Point Processes (DPP) from random matrix theory to select a diverse and uncorrelated subset of these rules, reducing redundancy. By rating data samples according to these selected rules and choosing the best ones for training, they enhance the performance of LLMs across various tasks, such as sentiment analysis, medical knowledge, mathematics, and code generation. The method is fully automated, reduces human bias, and is adaptable to different domains, leading to better-trained models without manual intervention.

**Strengths:**

•	The authors use DPP to select rules that aren’t too similar, enhancing rule diversity. This approach enables the rules to assess data from different perspectives, improving data selection and ultimately boosting model performance.

•	They show that rule correlation value has a positive Pearson correlation with MSE, supporting the value of diverse rule selection. Furthermore, by comparing the results to randomly selected rules, they demonstrate that DPP can effectively identify these diverse rules, resulting in improved MSE.

•	The models trained with their selected data outperformed those trained with data chosen by other methods, such as DSIR, QuRating, and Random 10 Rules generated by GPT, demonstrating the effectiveness of their approach and using DPP for rule selection.

•	Their method is fully automated, removing the need for human-made rules, which makes it adaptable to various tasks. This automation allows it to be easily applied across different situations and domains, providing a flexible approach to data selection.

**Weaknesses:**

•	The reliance on LLMs like GPT for rule generation raises concerns about potential selection biases. Biases in the generated rules could lead to an overrepresentation or underrepresentation of certain types of data, which may impact the fairness and effectiveness of the data selection process.

•	The authors are encouraged to test their framework on larger models within the same family, such as LLaMA2-7B and LLaMA2-13B, to provide further evidence of its scalability and effectiveness.

•	There are no experiments that single out DPP's effectiveness, leaving it unclear how much of the performance can be attributed to the LLM's capabilities versus DPP's role. Without a baseline comparison using only GPT-4 for rule generation, the specific contribution of DPP remains uncertain, making it difficult to gauge its true impact within the framework.

**Questions:**

•	The authors should address potential selection biases introduced by GPT in rule generation, as biases in the generated rules could lead to an overrepresentation or underrepresentation of certain types of data. One way to assess this is by comparing the distribution of data selected using their method with the original data distribution. Additionally, comparing this distribution with that of baseline methods would provide insights into how their approach stands out in terms of bias mitigation.

•	The authors could consider using GPT-4 to perform DPP’s role by adding a refinement step that prompts GPT-4 to filter out and select a subset of r rules that are both highly relevant to the task and diverse, using a well-constructed prompt to achieve this. This refined subset could serve as a baseline for evaluating the effectiveness of the DPP alone. If the authors have already explored a similar approach, it would be helpful to clarify this and provide any reasons for choosing not to pursue it.

---

> ### Author Response · Authors · 2024-11-18
> **Response to Reviewer 1T3n**
>
> We appreciate the valuable feedback and below are our responses (all the line numbers and sections are referred to the new pdf version):
>
> **1. Reliance on GPT for rule generation:**
>
> We thank the reviewers for the insightful comments! To maintain interpretability, our rules are set to be natural language forms. The reliance on power LLM (and the most common one is GPT) for such text rules is quite reasonable. In order to show that the GPT rules are generally reliable and unbiased, we have provided the **comparison against using Claude as the rule generator in Section A.6.5**(Line 988-1018). We use PCA to visualize the embedding (generated using sentence-transformer) of the two sets of rules. It is observed that the two clusters of rules generated by GPT and Claude completely overlap, demonstrating no distinct separation. This suggests that GPT-4 functions effectively as an unbiased rule generator. We appreciate the reviewer’s suggestion and hope adding this comparison experiment would help resolve some of the concerns on the bias of GPT.
>
> **2. Models within same family:**
>
> For previous such as QuRating[1], only one model architecture was used for the experiments. In our case, due to computing resource constraints and the comprehensive scope of our experiments which cover multiple domains and baseline methods, we were not able to repeat the finetuning experiments for the Llama2 family. However, in our case, we considered both Pythia and Llama. Moreover, in experiments in Section 4 (Evaluation A), we have explored the variation in the model size and we **included the results for both Llama3-8B and Llama3-70B (please see Line 362-372 and Section A.6.4, Line 951-986)**. We observe very similar phenomena and this indicates that our results are generally applicable regardless of the model architecture. Currently, due to computing resource constraints and the comprehensive scope of our experiments which cover multiple domains and baseline methods, we are not able to repeat the experiments for further models. For the future version, we will try to add experiments for model architectures such as the Llama2 family.
>
> **3. (a) Ablation of DPP, and (b) Compare with GPT 10 uncorrelated rules:**
>
> (a) Within our framework, we compared setups using Random 10 rules and DPP 10 rules as part of an **ablation study to assess the effectiveness of DPP. Please see Line 354-361.**
>
> (b) Following the reviewer’s suggestion, we also tested GPT-generated 10 uncorrelated rules to contrast with rule generation solely by GPT-4. Before, we mentioned in Section A.7.9 (Lines 1343-1409), we analyzed the correlation of these GPT-generated uncorrelated rules and found that, despite claims of being “uncorrelated,” their actual rating scores showed significantly higher correlation compared to the DPP method. This underscores the value of our Rule Correlation measure for quantitatively assessing correlations. We have highlighted this point to our readers in Section 5.3 (Lines 485-491). Now, we **added new benchmark results presented in Tables 1-3 (Lines 458-525) confirm that our DPP method outperforms the GPT-Uncorrelated rules**. We are grateful to the reviewer for this insightful suggestion!
>
> **4. Data distribution:**
>
> As suggested, we have **included an analysis of the data distribution in Section A.7.7 (Lines 1323-1335 and Figure 10)**. Using the Code domain as an illustrative example, we demonstrated that DPP tends to select more domain-related data, such as from GitHub and StackExchange.
>
> We are grateful to reviewer 1T3n for these valuable suggestions! We hope that these additional discussions and experiments in the new version will encourage reviewer 1T3n to consider a higher score!
>
> [1] Alexander Wettig, Aatmik Gupta, Saumya Malik, and Danqi Chen. Qurating: Selecting high-quality
> data for training language models. arXiv preprint arXiv:2402.09739, 2024.

---

> ### Comment · Reviewer_1T3n · 2024-11-20
>
> **1. Reliance on GPT for rule generation:**
>
> The selection bias in foundation models might be similar since they are trained on comparable data. However, upon reviewing Section A.6.5 and the PCA visualization (Figure 8), there doesn’t seem to be a clear overlap between the rule clusters produced by GPT and Claude, as described. It would be helpful if authors clarified how they are measuring this overlap and provided additional details on the metrics or methods used to support this conclusion.
>
> **2. Models within the same family:**
>
> While the Pearson correlation between rule correlation and MSE is not as high as 0.6 as mentioned in the paper(line 362-372 and figure 2), we can still observe that as rule correlation increases, MSE also increases due to the positive Pearson correlation.
>
> **3. GPT 10 uncorrelated rules:**
>
> It seems the approach taken does not fully align with the proposed method. The authors could consider using GPT-4 to replicate DPP's role by adding a refinement step where GPT-4 filters and selects a subset of r rules that are both highly relevant to the task and diverse. The resulting subset could then serve as a direct baseline for evaluating the effectiveness of DPP in comparison.
>
> **4. Data distribution:**
>
> DPP tends to select domain-relevant data. To better understand the data distribution, the authors could analyze the distribution of the original dataset and the distribution of selected samples resulting from different selection methods, including DPP. Metrics such as the length of training samples, cross-entropy loss before fine-tuning, and cyclomatic complexity for coding datasets could provide insights.
>
> The authors have addressed my comments partially so I will maintain my current score.

---

> ### Author Response · Authors · 2024-11-21
> **Response to Comment of Reviewer 1T3n**
>
> We appreciate the reviewer for the reply and new insights!
>
> **1. Reliance on GPT for rule generation:**
>
> We have further **computed the Wasserstein distance as a quantitative evaluation of the distribution differences**. Specifically, we compute the distance between the GPT rules and Claude rules. Then we randomly split GPT rules into two parts and computed their Wasserstein distance and similarly for the Claude rules. By comparing these values we found no clear distribution bias when switching from one rule generator to the other. Please see Table 4 (Line 1022-1032 of the most recent version).
>
> Moreover, to address the concern about the general bias of LLMs, in Line 1034-1042, compared to human-generated rules, we **prompt GPT-4 to generate ethical and safety rules and compare them with the human-designed rules in Collective Constitutional AI[1]**. We asked 3 authors who have not seen the constitutions to distinguish the rules blindly. We get an average accuracy of 20.7%, suggesting it is indeed hard to distinguish between rules generated by GPT and those designed by humans. All these discussions underscore the potential of GPT as a reliable rule generator that **is capable of producing rules that are comparable to those crafted by human experts**. We thank the reviewer for pointing this out and we hope this in some aspect addresses the reviewer’s concern (and maybe some readers’ concerns) about rule generation bias.
>
> **2. Models within the same family:**
>
> We agree that the Pearson correlation varies based on different domains/tasks, and indeed positive Pearson correlation is shown between MSE and RuleCorrelation, indicating that rule correlation indeed reflects the rule quality in certain aspects.
>
> **3. GPT 10 uncorrelated rules:**
>
> We understand that the reviewer would like to test the exact ability of DPP to select out uncorrelated rules, compared to GPT selection. We have now **further added experiments for all the domain fine-tunings. The results are shown in Tables 21, and 22 (Line 1574-1590)**. We note that it generally underperforms compared to our methods. Moreover, we calculated the rule correlation again, and found it is a bit smaller than random 10 rule selections, but still generally higher than DPP (the rule correlation and rule indices are included in Table 20, Line 1566-1572). We thank the reviewer for this suggestion and our new experiments validate the important role of DPP in rule selection.
>
> **4. Data distribution:**
>
> We appreciate the detailed suggestions for analyzing the data distribution. We have provided a deeper exploration of the data distribution in A.7.7 (Line 1379-1479). For **Code domain, we studied the metadata distribution** and it shows DPP selects more GitHub and StackExchange data. (We also studied the text lengths but found no clear pattern for the Code data. Here we did not use the cyclomatic complexity because due to our selection of pretrain data, even if the texts contain some code, they also contain statements or questions.) Moreover, we added an exploration of the IMDB domain. First from the **text length distribution**, we see DPP generally selects longer data. Then we studied the **bigram entropy distribution**, which indicates diversity of texts, and found that DPP selects texts with higher entropy/diversity.
>
> Again we thank the reviewer for the active discussion and insightful suggestions! We hope these new experiments would help address the rest of the reviews’s concerns and we would appreciate a raise of the score!
>
> [1] Saffron Huang, Divya Siddarth, Liane Lovitt, Thomas I Liao, Esin Durmus, Alex Tamkin, and Deep Ganguli. Collective constitutional ai: Aligning a language model with public input. In The 2024 ACM Conference on Fairness, Accountability, and Transparency, pp. 1395–1417, 2024.

---

> > ### Author Response · Authors · 2024-11-30
> > **Friendly Reminder**
> >
> > We thank the reviewer once again for their valuable comments! As the deadline approaches, we sincerely hope the reviewer could take a look at our updates, if possible.  We tried all possible ways to find GPU resources in a limited time and finish the experiments as soon as we could, with the hope of getting further valuable feedback from the reviewer! We sincerely hope that these new experiments and analyses address the reviewer’s concerns!

---

> > > ### Author Response · Authors · 2024-12-02
> > > **Friendly Reminder (1 day remaining for discussion)**
> > >
> > > As there is only 1 day remaining in the discussion period, we would like to kindly ask the reviewer if there are any further concerns. We sincerely thank the reviewer for their valuable comments and active participation in the discussion. We hope that the additional experiments and analyses have effectively addressed the previous concerns and if possible, we would greatly appreciate a reconsideration of the final score. Once again, we are truly grateful for the reviewer’s efforts and thoughtful feedback throughout the entire rebuttal period!

---

> > ### Author Response · Authors · 2024-12-03
> > **Rebuttal period ending**
> >
> > Dear Reviewer 1T3n,
> >
> > We have added GPT 10 uncorrelated rules experiments and analysis for the data distribution. Since the rebuttal period is closing, can you please provide the last response? We would be grateful for it!

---

### Official Review · Reviewer_94XN · 2024-11-03

**Soundness:** 2
**Presentation:** 3
**Contribution:** 2
**Rating:** 5
**Confidence:** 3

**Summary:**

This paper proposes a rule-based framework to rate and select samples for LLM training. The framework first generates a set of rules using GPT-4. Then, it uses the determinant point process (DPP) to select the most diverse subset from the previous set of rules based on the score vectors of a batch of randomly samples data. The selected rules are then used to rate and select data for LLM training. Experiments show the effectiveness of the proposed framework to some extent.

**Strengths:**

1. The authors are the first to introduce the mathematical rule evaluation metric, which is interesting and novel.
2. The motivation of  exploring DPP sampling for rules selection is technical sound.
3. The authors provide extensive experiments span various settings, including general pre-training and domain-specific fine-tuning in fields such as IMDB, Medical, Math, and Code
4. The paper is well-written and easy to follow.

**Weaknesses:**

1. The necessity of using an automated method to select some rules from the rule set is questionable, especially when the total number of rules is not large (N=50) and each rule is not long (see Tables 4, 5, and 13-16). In this case, the cost of manual rule selection is totally acceptable and manual selection is more reliable than automatic methods.
2. According to the experimental results in Table 1-3, the performance improvement achieved through DPP sampling appears to be somewhat limited. The authors may want to report the mean and standard deviation of the performance of different methods in multiple runs, and also provide a significance test.
3. Some important baselines are missing, such as LESS (ICML 2024) [1], IFD (ACL 2023) [2], SelectIT [3], DiverseEvol [4], ZIP [5], and InsTa [6]. The author should select as least two of latest data selection methods for comparison.
4. In Section A.6.6, the authors generate 10 uncorrelated rules using GPT-4 for Code and Math domains. I would like to suggest the author to add “GPT 10 Uncorrelated Rules” as a baseline in experiments in Section 5 to see the performance of models based on “GPT 10 Uncorrelated Rules”.
5. Some important analysis that could provide better insight to readers is missing as follows. (1) Why does “All 50 Rules” underperforms “DPP 10 Rules”? As claimed in Lines 489-493, the diversity in the data rating step can improve the model performance. According to this claim, the performance of “All 50 Rules” should exceed that of “DPP 10 Rules”, as the diversity of all 50 rules is obviously greater than that of the 10 selected rules. (2) What are the differences in the distribution of samples selected based on the selected rules?

[1] LESS: Selecting Influential Data for Targeted Instruction Tuning

[2] From Quantity to Quality: Boosting LLM Performance with Self-Guided Data Selection for Instruction Tuning

[3] SelectIT: Selective Instruction Tuning for Large Language Models via Uncertainty-Aware Self-Reflection

[4] Self-Evolved Diverse Data Sampling for Efficient Instruction Tuning

[5] Entropy Law: The Story Behind Data Compression and LLM Performance

[6] Instruction Matters: A Simple yet Effective Task Selection for Optimized Instruction Tuning of Specific Tasks.

**Questions:**

Please see the weaknesses for the details.
Overall, it is a good paper with some flaws. I will consider raising my score if the authors can effectively address my concerns.

---

> ### Author Response · Authors · 2024-11-18
> **Response to Reviewer 94XN (Part 1/2)**
>
> We appreciate the valuable feedback and below are our responses (all the line numbers and sections are referring to the new pdf version):
>
> **1.Automatic v.s. Manual rule selection:**
>
> Using 50 rules is simply due to GPU resource constraints. We prioritized demonstrating the general applicability of our method by covering many domains. In fact, there are potentially hundreds of aspects to evaluate data quality[1][6] (for example [6] uses 133 rules). Also, our completely automated method is an advantage compared to rule selection by humans, where **human intervention is needed again for each new task, making it costly and challenging to integrate into certain pipelines**, such as composite reward models in RLHF. Moreover, **varying preferences among individuals can inevitably introduce bias** during rule selection. Additionally, human selection of uncorrelated rules is generally based on the **rules' semantic meanings**, but rules can show different rating correlations on different datasets, resulting in bias and redundancy on data rating and also hindering customization or re-weight of rules. In contrast, we choose to select orthogonal rules based on their **score representations, which are guaranteed to be data-dependent**. We have added a discussion in Section 5.3 (Line 485-491). We thank the reviewer for pointing this out!
>
> **2. (a) Performance improvement  and (b) Multiple trials:**
>
> (a) Our GPU resources are not as extensive as groups like QuRating, which are able to pretrain LLMs. In contrast, we perform our fine-tuning with only 20K samples, yet we still achieve significant improvements and superior performance over baselines. For instance, in Medical domain, our DPP method **improved the average accuracy of Pythia by about 5%**, and it outperformed other baselines, such as achieving a **7.2% increase in accuracy over DSIR**[5].
>
> (b) We added the results to be **3 trials for Uniform Sampling and Random 10 Rules and reported their mean and standard deviation in Section A.7.3** (Line 1198 - 1227). Due to limited GPU resources, we were unable to perform multiple trials for all experiments, but it is quite common in LLM research to report one trial now, for example see [1][3][4].
>
> **3. More baselines:**
> We have invested significant efforts to implement the suggested methods. Due to the time-consuming nature and unreliable codes of some methods, we eventually reproduced **LESS and DiverseEvol. We added their results for all the 5 settings in Table 7-9** (Line 1163-1195)  to further validate the advantages of our methods. We appreciate the reviewer’s suggestion and mention of these baseline methods!

---

> ### Author Response · Authors · 2024-11-18
> **Response to Reviewer 94XN (Part 2/2)**
>
> **4. GPT 10 uncorrelated rules:**
>
> We thank the reviewer for this great suggestion! The **experiments for GPT 10 uncorrelated rules for all 5 settings are now added to Table 1-3** (Line 456-525). By comparing against this, the advantage of our methods is more obvious. Moreover, this is directly prompting GPT to generate 10 uncorrelated rules. We also added experiments to prompt GPT to select 10 uncorrelated rules from a 50-rule pool. Please see A.7.10 (Line 1528-1559) for the results.
>
> **5. (a) All 50 rules v.s. DPP 10 rules, and (b) Data distribution:**
>
> (a) In Section 4, we have demonstrated multiple plots showing that the **optimal r is not near the boundaries (Figures 3c, 3f)**. The non-monotonicity in r is itself an interesting result reported by our paper. We conjecture that orthogonality is important: **larger r covers more aspects but may introduce correlations that disrupt the balance or evenness, causing the rating biased toward certain aspects**. When we mention the “diversity” brought by orthogonality, we mean under the condition of balance, or essentially a “balanced diversity” (Line 496). We appreciate the reviewer posting such an insightful question which highlights an interesting crucial part of our work. In fact, one of the contributions of our paper is to explore the reason behind the optimal r using rule correlation/orthogonality, proposing quantitative metrics and various experiments to support our hypothesis. We have **added some discussion about these in Line 493-499**.
>
> (b) We have examined the data distribution and added discussion in Section A.7.7 (Lines 1352-1450 and Figure 10,11,12), using the Code and IMDB domains as illustrative examples. We demonstrated that DPP tends to select more domain-related data, longer texts, and texts with higher diversity (quantified using bigram entropy).
>
> In summary, we appreciate reviewer 94XN for so many insightful questions and suggestions!  We hope that our additional experiments and discussion bring more clarification to the concerns and would really appreciate if reviewer 94XN could raise the score.
>
> [1] Together AI. Red pajama. Blog post on Together AI, 2023. https://www.together.ai/
> blog/redpajama.
>
> [2] Alexander Wettig, Aatmik Gupta, Saumya Malik, and Danqi Chen. Qurating: Selecting high-quality
> data for training language models. arXiv preprint arXiv:2402.09739, 2024.
>
> [3] Weizhe Yuan, Richard Yuanzhe Pang, Kyunghyun Cho, Sainbayar Sukhbaatar, Jing Xu, and Jason Weston. Self-rewarding language models. arXiv preprint arXiv:2401.10020, 2024
>
> [4]Zhiqing Sun, Yikang Shen, Qinhong Zhou, Hongxin Zhang, Zhenfang Chen, David Cox, Yiming Yang, and Chuang Gan. Principle-driven self-alignment of language models from scratch with minimal human supervision. Advances in Neural Information Processing Systems, 36, 2024.
>
> [5] Sang Michael Xie, Shibani Santurkar, Tengyu Ma, and Percy S Liang. Data selection for language models via importance resampling. Advances in Neural Information Processing Systems, 36, 2024.
>
> [6] Huang, Saffron, et al. "Collective Constitutional AI: Aligning a Language Model with Public Input." The 2024 ACM Conference on Fairness, Accountability, and Transparency. 2024.

---

> ### Author Response · Authors · 2024-11-24
> **Gentle Reminder**
>
> We sincerely appreciate the insightful comments provided by Reviewer 94XN! Following the reviewer’s valuable suggestions, we have conducted the recommended experiments and added the corresponding discussions. This is a friendly reminder that the deadline of the rebuttal period is approaching.  We would really appreciate it if the reviewer could take a look at our responses at their earliest convenience. We hope that the additional experiments and discussions address the reviewer’s concerns effectively and would be deeply grateful if the reviewer could consider raising the score.

---

> > ### Comment · Reviewer_94XN · 2024-11-25
> >
> > Thank you for providing a detailed response along with additional experiments and explanations in the revised version. However, my main concerns remain unaddressed.
> >
> > The performance improvement of DPP 10 Rules compared to GPT-Uncorrelated and Random 10 Rules is not significant. For instance, in Table 1, the average performance of GPT-Uncorrelated, Random 10 Rules, and DPP 10 Rules are **42.6**, **42.8**, and **43.2**, respectively. The performance enhancement provided by DDP 10 Rules lacks both statistical significance and practical value.
> >
> > The improvement of **1.3** in the medical domain and **1.2** in the math domain compared to GPT-Uncorrelated suggests that the performance improvement is insufficient to demonstrate the DPP 10 rules as a viable and practical solution.
> >
> > Therefore, I maintain a score of 5.

---

> > > ### Author Response · Authors · 2024-11-26
> > >
> > > We really appreciate the response! Based on the reviewer’s feedback, we have **added the experiments including multiple trials for various methods (please see A.7.3 Tables 11 and 12, Line 1129-1247, we also updated the averaged results in Tables 1 and 2.)** As suggested by the reviewer, we also conducted a significant test: **we performed Welch’s t-test to compare DPP 10 Rules with various other methods. Most of the p values are around 1e-3 or 1e-4. For the detailed t-statistics and p-values, please see A.7.3 Tables 13 and 14, Line 1251-1270.** This analysis confirms the significance of the advantage of our method.
> > >
> > > Below are some further discussion:
> > >
> > > 1. We would like to emphasize that our data source is a pre-train dataset (SlimPajama) and the selected data size is only 2% of it. Having such large domain improvements using pre-train data is already noteworthy. As a comparison, in DSIR [1] (one of the most popular data selection baselines), the performance improvement of DSIR compared with its baseline methods are the following: 1.18 over uniform sampling, 0.3 over manual curation, 0.88 over heuristic classification, and 0.11 over top-k heuristic classification (see Table 1 of their paper). In QuRating [2], they have 0.8 improvement over the uniform sampling. In our case, the advantage of DPP compared to other baseline methods is substantially more significant. For example, compared to uniform sampling, **we have an improvement 9.6 on IMDB, 2.7 on Medical, 2.6 on Math, and 9.9 on Code**.
> > >
> > > 2. Moreover,  the **baseline GPT-Uncorrelated is essentially a variant of our framework**. It still uses our automated orthogonal-rule-based pipeline but only replaces the DPP part by GPT. Even compared to GPT-Uncorrelated, **DPP 10 Rules still has noticeable improvements across all domains. For example, a 5.4 improvement on Code.**
> > >
> > > 3. Besides the benchmark performance, we want to highlight that to our best knowledge, **we are the first to systematically study the fine-grained rule-based approach for quality data selection, the first to propose measures for evaluating quality metrics (rules), and the first to propose the DPP algorithm to achieve this.** Our pipeline is completely human-free and achieves effective high-quality data selection while maintaining great quality interpretability.
> > >
> > > 4. Our methodology is very general; it is broadly applicable to settings such as pretraining, fine-tuning, and various other settings of data selection. It could also be used for composite reward models in RLHF [3].
> > >
> > > **Following the reviewer’s suggestions, we have conducted significant additional experiments, including experiments for more data selection baseline methods, multiple trials for various methods, and experiments for GPT uncorrelated rules.** We have used all the GPU resources we can find in a limited time, and have tried our best efforts to provide updates promptly for active discussion. We hope these address the reviewer’s concern. We are grateful for your insightful feedback and suggestions to make our paper more robust! **Based on the additional experiments and analysis, we sincerely hope you can consider a re-evaluation of our work and consider a raise of the score!** Thank you once again for your valuable comments and active participation in the discussion!
> > >
> > > [1] Xie, Sang Michael, et al. "Data selection for language models via importance resampling." Advances in Neural Information Processing Systems 36 (2023): 34201-34227.
> > >
> > > [2] Wettig, Alexander, et al. "Qurating: Selecting high-quality data for training language models." arXiv preprint arXiv:2402.09739 (2024).
> > >
> > > [3] Wang, Zhilin, et al. "HelpSteer2: Open-source dataset for training top-performing reward models." arXiv preprint arXiv:2406.08673 (2024).

---

> > > ### Author Response · Authors · 2024-11-27
> > >
> > > We sincerely appreciate Reviewer 94XN again for providing constructive suggestions and active engagement in the discussion. As today is the last day to submit a revised pdf, we want to kindly follow up to see if there are any remaining concerns or questions that we can address. We would be grateful if you could consider raising the score after seeing our new experiments and analysis! Wishing you a happy thanksgiving!

---

> > > > ### Comment · Reviewer_94XN · 2024-11-28
> > > >
> > > > I maintain that this paper does not satisfy the criteria for acceptance. Reasons are below.
> > > >
> > > > 1. Training with 20M tokens (2% of 1B general data)  for CPT on LLaMA3-8B is meaningless, as llama3 utilize ~15T tokens for pre-training. It is difficult to envision how CPT training with 20M tokens (for llama3) can enhance performance in general tasks, like code and math.
> > > >
> > > > 2. An improvement of one point over the baseline is insignificant in LLMs, even if DSIR has been accepted.
> > > >
> > > > 3.  Table 1 demonstrates the performances in the most important and commonly used benchmarks. The performance gaps between models trained using different data selection methods typically fall within 0.5-point. This variance is largely attributed to the inherent randomness of LLM inference and holds no practical significance.
> > > >
> > > > 4. The performace gap between GPT-Uncorrelated and DDP 10 rules in important to evaluate the contributions of this paper.

---

> > > > > ### Author Response · Authors · 2024-11-28
> > > > >
> > > > > Thanks for the continued feedback.
> > > > >
> > > > > 1. As an example, the LIMA paper [1] by Meta AI demonstrated that fine-tuning a 65B Llama model with only 1K high-quality examples resulted in remarkable performance. **Compared to our 20M examples, 1K is tiny. Moreover, when compared to the pretraining data of the 65B Llama model, 1K is negligible!** Would this also be considered meaningless? The LIMA paper is just one among many [1][2][3][4][5][6][7][8][9][10][11][12][13][14][15] that highlight the critical importance of data quality, even when the dataset is small.
> > > > >
> > > > > Furthermore, **when fine-tuning LLMs, the size of the fine-tuning dataset is typically much smaller than the pretraining dataset. The difficulty of easily predicting the influence of fine-tuning data does not imply that this influence does not exist.** If research outcomes were always predictable and can be pre-“envisioned”, there is no purpose for conducting research.
> > > > >
> > > > > 2. **We completely disagree with the statement that 1 point improvement is insignificant in LLMs.** Could the reviewer kindly provide supporting evidence or examples to substantiate this claim?
> > > > >
> > > > > 3. As discussed clearly in the paper, we anticipated that improvements on general benchmarks would be smaller compared to domain-specific fine-tuning. Nevertheless, **we presented results from four domains that demonstrate obvious and significant improvements.** We are not sure why the reviewer appears to overlook these results now.
> > > > >
> > > > > 4. There might be a misunderstanding regarding the primary contribution of our paper: to our best knowledge, **we are the first to systematically study the fine-grained rule-based approach for quality data selection, the first to propose measures for evaluating quality metrics (rules), and the first to propose the DPP algorithm to achieve this.** We agree that the performance gap between GPT-Uncorrelated and DPP 10 Rules is one of the evaluators. This is why **we have provided the significant test to compare DPP with GPT-Uncorrelated**, which is also strongly requested by the reviewer. We do not understand why the reviewer is ignoring the significance test now.
> > > > >
> > > > > We hope this response clarifies our contributions and addresses the reviewer’s concerns. We would sincerely appreciate additional support or clarification for the claims made, and we hope this exchange reduces any remaining confusion. Thank you for your time and consideration.
> > > > >
> > > > > [1] Zhou, Chunting, et al. "Lima: Less is more for alignment." Advances in Neural Information Processing Systems 36 (2024).
> > > > >
> > > > > [2] Xia, Mengzhou, et al. "Less: Selecting influential data for targeted instruction tuning." arXiv preprint arXiv:2402.04333 (2024).
> > > > >
> > > > > [3] Alexander Wettig, Aatmik Gupta, Saumya Malik, and Danqi Chen. Qurating: Selecting high-quality data for training language models. arXiv preprint arXiv:2402.09739, 2024.
> > > > >
> > > > > [4] Sang Michael Xie, Shibani Santurkar, Tengyu Ma, and Percy S Liang. Data selection for language models via importance resampling. Advances in Neural Information Processing Systems, 36, 2024.
> > > > >
> > > > > [5] Li, Ming, et al. "From quantity to quality: Boosting llm performance with self-guided data selection for instruction tuning." arXiv preprint arXiv:2308.12032 (2023).
> > > > >
> > > > > [6] Pang, Jinlong, et al. "Improving Data Efficiency via Curating LLM-Driven Rating Systems." arXiv preprint arXiv:2410.10877 (2024).
> > > > >
> > > > > [7] Cao, Yihan, et al. "Instruction Mining: Instruction Data Selection for Tuning Large Language Models." arXiv preprint arXiv:2307.06290 (2023).
> > > > >
> > > > > [8] Wei, Lai, et al. "Instructiongpt-4: A 200-instruction paradigm for fine-tuning minigpt-4." arXiv preprint arXiv:2308.12067 (2023).
> > > > >
> > > > > [9] Wu, Shengguang, et al. "Self-evolved diverse data sampling for efficient instruction tuning." arXiv preprint arXiv:2311.08182 (2023).
> > > > >
> > > > > [10] Chen, Hao, et al. "Maybe only 0.5% data is needed: A preliminary exploration of low training data instruction tuning." arXiv preprint arXiv:2305.09246 (2023).
> > > > >
> > > > > [11] Hoffmann, Jordan, et al. "Training compute-optimal large language models." arXiv preprint arXiv:2203.15556 (2022).
> > > > >
> > > > > [12] Zhao, Wayne Xin, et al. "A survey of large language models." arXiv preprint arXiv:2303.18223 (2023).
> > > > >
> > > > > [13] Jang, Joel, et al. "Exploring the benefits of training expert language models over instruction tuning." International Conference on Machine Learning. PMLR, 2023.
> > > > >
> > > > > [14] Wang, Yizhong, et al. "Self-instruct: Aligning language models with self-generated instructions." arXiv preprint arXiv:2212.10560 (2022).
> > > > >
> > > > > [15] Ivison, Hamish, et al. "HINT: Hypernetwork Instruction Tuning for Efficient Zero-& Few-Shot Generalisation." arXiv preprint arXiv:2212.10315 (2022).

---

> > > > > > ### Comment · Reviewer_94XN · 2024-12-03
> > > > > >
> > > > > > Thank you for your further explanation!
> > > > > >
> > > > > > The authors should note that instruction tuning (SFT) and CPT are entirely different concepts. SFT is designed to help a model to understand different instructions and learn the desired output style/format, without needing of knowledge injection. On the other hand, the CPT phase is meant for injecting new knowledge. Therefore, a large-scale dataset (preferably with data not extensively encountered during the pre-training phase) is required for CPT.  Therefore, using a small amount of high-quality data for SFT is reasonable, while using 20M data for CPT is not very meaningful.
> > > > > >
> > > > > > The output of a large model inherently exhibits randomness. Changing the inference hyperparameters or using different versions of transformers/VLLM packages may result in a 1-2 point difference in evaluation.  Therefore, we believe that a 1 point improvement is not significant in practice.

---

> > > > > > > ### Author Response · Authors · 2024-12-03
> > > > > > >
> > > > > > > Thanks for the reply. First, we would greatly appreciate it if the reviewer can provide support to the statements instead of making standalone claims. Moreover, the reviewer's discussion focus is somewhat biased and not related to the main focus or contribution of our paper. For example, we have conducted fine-tuning in multiple domains but somehow the results are neglected by the reviewer.

---

> > > > > ### Author Response · Authors · 2024-12-02
> > > > > **Friendly Reminder (1 day remaining for discussion)**
> > > > >
> > > > > As there is only 1 day remaining in the discussion period, we would like to kindly ask the reviewer if there are any further concerns. We sincerely thank the reviewer for their valuable comments and active participation in the discussion. We hope that the additional experiments and analyses have effectively addressed the previous concerns and if possible, we would greatly appreciate a reconsideration of the final score. Once again, we are truly grateful for the reviewer’s efforts and thoughtful feedback throughout the entire rebuttal period!

---

> > > > > ### Author Response · Authors · 2024-12-03
> > > > > **Rebuttal period ending**
> > > > >
> > > > > Dear Reviewer 94XN,
> > > > >
> > > > > We have added the significant tests, and provided analysis and discussion about it. Since the rebuttal period is closing, can you please provide the last response? We would be grateful for it!

---

### Meta-Review · Area_Chair_UZ6i · 2024-12-20

**Metareview:**

This paper proposes an automated, rule-based framework for selecting high-quality LLM training data without human intervention. The method leverages a determinantal point process to select diverse and independent rules by evaluating the orthogonality of score vectors, thereby enhancing rule diversity and improving data selection. The framework is fully automated, adaptable to various tasks, and eliminates human bias, offering an efficient and flexible approach for data selection in LLM training.

However, despite multiple rounds of discussion between the authors and reviewers, the paper still presents two key concerns:

**Lack of multiple random trials.** Different LLMs may exhibit variability in their results due to differences in transformer versions or inherent randomness in the inference sampling strategies. This randomness can lead to inconsistencies in the results across multiple runs. Given that the performance improvements shown in Table 1 are relatively modest (with improvements of less than 0.5 compared to the second-best method), it remains unclear whether the reported gains are genuinely due to the proposed method or if they are influenced by the randomness of the inference process.

**Absence of analysis on varying sample proportions.** In real-world scenarios, the scale of data for Continual Pre-Training can fluctuate significantly. Specifically, the proportion of data used for Continual Pre-Training relative to the model's previous pre-training data can vary across different contexts. A robust data selection or sampling method should be capable of adapting to these varying scales. However, the paper lacks a discussion or experiments regarding how the proposed method performs under different data scales for Continual Pre-Training, which limits its practical applicability.

**Additional Comments On Reviewer Discussion:**

The paper receives mixed scores, with three weak positives (6, 6, 6) and one negative (5).

Reviewer 94XN raises concerns about the scale of data for Continual Pre-Training.

Reviewer 1T3n believes his concerns have been partially addressed and still raises issues regarding the measurement of rule overlap between GPT and Claude, the weak correlation between rule correlation and MSE, the alignment of GPT-4 with DPP for rule selection, and the need for better analysis of data distribution and selection methods.

Reviewer ZVjx and Reviewer 9u4d raise the score, and the authors have addressed their concerns during the rebuttal period.

AC agrees with Reviewer 94XN on the potential issues of the absence of analysis on varying sample proportions, and notes that the lack of multiple random trials is another aspect where the paper could be improved. Therefore, the final decision is to reject the paper.

---

### Decision · Program_Chairs · 2025-01-22

Reject